# A malware classification method based on directed API call relationships

**Cuihua Ma[1,2,3], Zhenwan Li[1], Haixia Long[1,2,3]\*, Anas Bilal[1]\*, Xiaowen Liu[1,2]**

**1** College of Information Science Technology, Hainan Normal University, Haikou, Hainan, China, **2** Hainan Engineering Research Center for Extended Reality and Digital Intelligent Education, Haikou, Hainan, China, **3** Key Laboratory of Data Science and Smart Education, Ministry of Education, Hainan Normal University, Haikou, Hainan, China

\* myresearch_hainnu@163.com (HL); a.bilal19@yahoo.com (AB)

**Data availability statement:** All relevant data is publicly available at (https://www.kaggle.com/datasets/8abfa8c0975603330354e778767d7f269cc4e3e707b6ad92c5ce22acb04ebf7c) and (https://github.com/ocatak/malware_api_class).

## Abstract

In response to the growing complexity of network threats, researchers are increasingly turning to machine learning and deep learning techniques to develop advanced models for malware detection. Many existing methods that utilize Application Programming Interface (API) sequence instructions for malware classification often overlook the structural information inherent in these sequences. While some approaches consider the structure of API calls, they typically rely on the Graph Convolutional Network (GCN) framework, which tends to neglect the sequential nature of API interactions. To address these limitations, we propose a novel malware classification method that leverages the directed relationships within API sequences. Our approach models each API sequence as a directed graph, incorporating node attributes, structural information, and directional relationships. To effectively capture these features, we introduce First-order and Second-order Graph Convolutional Networks (FSGCN) to approximate the operations of a directed graph convolutional network (DGCN). The resulting directed graph embeddings from the FSGCN are then transformed into grayscale images and classified using a Convolutional Neural Network (CNN). Additionally, to mitigate the effects of imbalanced datasets, we employ the Synthetic Minority Over-sampling Technique (SMOTE), ensuring that underrepresented classes receive adequate attention during training. Our method has been rigorously evaluated through extensive experiments on two real-world malware datasets. The results demonstrate the effectiveness and superiority of our approach compared to traditional and graph-based malware classification techniques.

## Introduction

Malware, short for "malicious software", is intentionally designed by individuals or organizations to damage or disrupt the normal functioning of endpoint devices. This category includes various types such as phishing scams, spyware, adware, viruses, rootkits, Trojans, worms, and ransomware [1,2]. The scope and scale of malware have surged in recent years, presenting an increasing challenge to cybersecurity. For example, the Verizon Business 2022 Data Breach Investigations Report revealed a 13% year-on-year increase in ransomware incidents in 2022,

**Funding:** This research was funded by the National Natural Science Foundation of China (No. 62262019 to H.L., No. 71762010 to X.L.), the Hainan Provincial Natural Science Foundation of China (No. 621QN241 to C.M., No. 823RC488 to H.L., No. 621RC1059 to X.L.), the Haikou Science and Technology Plan Project of China (No. 2022-016 to H.L.).

**Competing interests:** The authors have declared that no competing interests exist.

surpassing the cumulative growth of the previous five years [3]. Additionally, McAfee Labs' Threat Report for Q2 2020 reported a significant rise in malware activity, with the average number of threats per minute increasing by 44 to 419, marking a 12% surge [4].These statistics highlight the increasing volume and complexity of malware threats, emphasizing the urgent need for more advanced and effective malware detection methods.

In response to the growing complexity of network threats, security firms are deploying various tools, while researchers leverage machine learning(ML) and deep learning(DL) techniques to develop advanced models for addressing these threats [5–10]. As malware types and behaviors continue to diversify, the methods to detect and classify them have grown progressively more complex.

Malware analysis can be broadly categorized into two main approaches: static and dynamic analysis. Static analysis involves examining software without execution, extracting insights such as opcode sequences from disassembled binaries, control flow graphs, and more [11–13]. These static features are essential for detecting known malware. However, static analysis has limitations—techniques like obfuscation or packing enable malware to evade detection, diminishing its effectiveness.

In contrast, dynamic analysis involves executing malware in a controlled virtual environment to observe its real-time behavior. This method provides detailed data, including API calls, system calls, instruction tracing, registry modifications, and memory writes [14]. Our research focuses exclusively on dynamic analysis using API sequences, providing a robust detection method by evaluating the actual behavior of malware, rather than relying on static attributes like file properties or code structure. Static features are more easily manipulated by malware developers, while API sequences are more resistant to manipulation. Additionally, API sequences offer valuable insights into program behavior, essential for understanding and identifying malware malware [15–18].

Over the past decade, many researchers have utilized traditional machine learning techniques for malware detection and classification, particularly focusing on API sequences [19–22]. While these methods laid the foundation, they often oversimplify the process by converting API names into numerical values, leading to a limited representation of the API sequence features. This oversimplified approach can result in suboptimal detection and classification performance. In contrast, some researchers have explored rule-based methods for extracting features from API sequences. While effective, these methods rely heavily on domain-specific knowledge and often struggle with complex datasets, lacking the ability to generalize across diverse scenarios.

Recently, there has been a shift toward using deep learning for malware detection, yielding superior results compared to traditional machine learning techniques [23–25]. Deep learning offers several advantages, including automatic feature extraction, the ability to handle large datasets, and adaptability in addressing diverse malware behaviors. These methods frequently utilize word embeddings to represent API calls or employ recurrent neural networks (RNN) and their variants, such as Long Short-Term Memory (LSTM) or Gated Recurrent Units (GRU), to capture the temporal relationships within API sequences. Some studies also treat API sequences as one-dimensional time series, applying convolutional neural networks (CNN) for local feature extraction. However, existing literature often provides oversimplified representations of API sequences. While some studies recognize the temporal dynamics of API calls, they overlook the structural intricacies within these sequences, resulting in limited feature representation [26–28]. Others have considered the structure of API sequences but treated them as unordered, neglecting the inherent sequential relationships, which ultimately led to suboptimal classification performance [29–31].

In real-world scenarios, malware classification faces significant challenges, particularly class imbalance, where certain malware families are underrepresented in the dataset. This imbalance can lead to biased models that predominantly classify the majority class, potentially missing less frequent malware variants. Another challenge is detecting novel malware—variants that were not seen during training. Traditional machine learning models often struggle with this, as they depend heavily on learned patterns from training data.

To address these challenges, we propose a novel malware classification method that utilizes directed API call relationships. our approach uses the Synthetic Minority Over-sampling Technique (SMOTE), a widely recognized method for generating synthetic examples of underrepresented classes. By balancing the dataset, SMOTE ensures that the classifier receives a fair representation of all malware classes, leading to improved performance, especially for detecting rare or emerging malware types. Our method begins by constructing a directed graph from the ordered sequence of API calls, where the nodes represent API types and the edges capture the relationships between API method calls. To enhance these sequences, we incorporate comprehensive node attribute information. This API-directed graph captures both the structural and attribute information of the nodes while preserving the ordered relationships inherent in the API calls. Our method uses a directed graph representation of API sequences to encode the structural and sequential behavior of malware. This graph-based representation is more flexible and robust than traditional static feature-based methods, enabling the model to generalize more effectively to unseen malware variants.

To address the limitations of existing Graph Convolutional Networks (GCN), which struggle to capture sequential relationships between nodes, we propose a strategy that approximates directed graph convolutional networks using first-order and second-order graph convolutional networks. This strategy constructs first-order and second-order adjacency matrices to approximate the directed graph's adjacency matrix. These matrices are processed through the GCN to produce embeddings that integrate both attribute and ordered structural information of the API-directed graph.

Finally, the resulting graph embeddings are transformed into grayscale images, and malware classification is performed using a CNN. The primary contributions of this paper are as follows:

- We construct a directed graph from API sequences, which captures not only the directed relationships between API calls but also includes the structural information among APIs. Additionally, we enhance the API sequence representation by incorporating detailed attribute features for the API nodes.
- To address the limitations of existing GCN methods, which do not capture ordered relationships between nodes, we propose a strategy that approximates directed graph convolutional networks using first-order and second-order graph convolutional networks. This strategy creates first-order and second-order adjacency matrices to approximate the directed graph's adjacency matrix, which are used in GCN operations to generate first-order and second-order undirected graph embeddings of the API sequence. The embeddings are subsequently fused to approximate the computations of a directed graph convolutional network.
- To validate our method, we conducted extensive experiments on two datasets, comparing it with state-of-the-art baselines.The results show that our method achieves optimal classification performance.

The rest of the paper is organized as follows. We introduce the related work in Sect 2. In Sect 3, 'Materials and Methods,' we primarily provide definitions and principles of various

methods, along with a detailed explanation of the method we propose. In Sect 4, 'Results and Discussion,' we mainly provide explanations regarding the source of our dataset, experimental setups, model configurations, and showcase a multitude of experiments conducted. Our conclusion can be found in Sect 5.

## Related work

The Windows API, a collection of functions that define how software interacts with Microsoft libraries, is crucial for dynamic analysis. Each API call represents a specific operation performed by either malware or benign software during system execution. Therefore, analyzing the sequence of these API calls is essential for detecting and classifying malicious software. Significant progress has been made in studying malicious behavior through system or program API sequences over the years. Research in this field can be broadly classified into two approaches: one using traditional machine learning techniques and the other utilizing deep learning algorithms. The following sections explore these two methodologies in detail.

### Methods for malware detection or classification based on traditional machine-learning techniques

Rabadi et al. [32] proposed two novel approaches focusing on API parameters. The first method extracts the number of API calls as a single feature, while the second counts the number of API arguments. They employed five machine learning techniques—Support Vector Machine (SVM), Extreme Gradient Boosting (XGBoost), Random Forest (RF), Decision Tree (DT), and Passive Aggressive (PA)—for malicious behavior detection. However, the study did not consider the direct or indirect invocation relationships between API sequences, potentially affecting detection and classification accuracy.

Vu et al. [21] introduced a malware classification method based on the Rete algorithm, which generates rules during pattern matching. Rule-based approaches, such as this one, often require domain expertise to craft specific rules, limiting their applicability in diverse contexts and reducing effectiveness with large-scale datasets.

Pektaş et al. [33] developed a Windows malware classification model based on runtime behavior. Their method involves mining and performing n-gram searches on API sequences to identify behavior-based features of malware. They employed the voting expert algorithm to extract malicious patterns from API calls and applied online machine learning algorithms for classification. Their method achieved a remarkable 98% accuracy in malware classification. However, it did not account for the ordered relationships between API calls or the comprehensive structural information within the sequences.

Xue et al. [34] used various classifiers to detect malware based on API call features, finding that the MLP algorithm achieved the best detection performance, with an accuracy of 91%. However, this study focused solely on the usage features of API calls and did not explore the invocation relationships between APIs, limiting overall classification accuracy.

Alazab et al. [26] proposed three strategies for grouping API calls to enhance the detection of malicious Android applications: fuzzy group, risk group, and destructive group. Their findings revealed that malicious applications often invoke different sets of APIs compared to benign applications and frequently request dangerous permissions to access sensitive data. They tested five machine learning methods—RF, J48, Random Tree (RT), k-Nearest Neighbors (KNN), and Naïve Bayes (NB)—for detection, with RF achieving the best performance at 94.30%. However, their approach did not account for the invocation relationships between APIs or the resulting inter-API information, leading to a relatively limited feature set.

## Methods for malware detection or classification based on deep learning algorithms

Recent studies have increasingly employed deep learning algorithms based on graph representation learning for malware detection and classification.

Liu et al. [27] conducted a significant study where they extracted API sequences from malicious programs using a Cuckoo sandbox. They refined the sequences by removing redundant API calls and evaluated the performance of various neural network models, including Bidirectional Long Short-Term Memory (BLSTM), GRU, Bidirectional Gated Recurrent Unit (BGRU), LSTM, and SimpleRNN, using a dataset of 21,378 samples. Their results showed that BLSTM achieved the highest accuracy in malware detection, with a performance of 97.85%. However, this method did not account for the invocation relationships and structural information between APIs, limiting the feature set's richness and potentially impacting detection accuracy.

Cai et al. [29] explored the local maliciousness of malware and developed an anti-interference detection framework using API fragments. They used an LSTM model to classify these fragments and employed ensemble learning to determine the overall API sequence classification. Tested on the Ali-Tianchi competition API database, their method achieved a notable accuracy of 97.34%. However, their approach treated the ordered relationships between API calls as unordered, causing a loss of essential information during feature extraction.

Karbab et al. [28] introduced MalDozer, an automated framework for Android malware detection and family attribution using deep learning techniques for sequence classification. MalDozer learns patterns from actual samples to detect Android malware, achieving an F1-Score between 96% and 99%, with a false positive rate of 0.06% to 2%. However, their method used a fixed-length approach for API sequence extraction, leading to information loss and scalability issues due to the need for manual length adjustments.

Zhang et al. [30] proposed an end-to-end model, SDGNet, which classifies directed graphs generated from sandbox API structure sequences. This model achieved an accuracy of 97.3%, extensively utilizing API invocation relationships to construct directed graphs and extracting graph features through various normalization techniques. However, the model primarily focused on first-order neighbor information, neglecting higher-order relationships.

Gao et al. [35] developed an Android malware detection and classification method using a graph convolutional network (GCN). By mapping applications and Android APIs onto a large heterogeneous graph and treating the task as a node classification problem, their system, GDroid, detected 98.99% of Android malware with a false positive rate of less than 1%.

Oliveira et al. [31] proposed a novel malware detection method using a deep graph convolutional neural network (DGCNN) that learns from both the API sequence and its associated behavior graph. The method achieved a high malware detection accuracy of 92.44%. However, their approach to transforming API sequences into a graph structure and using DGCNN resulted in limited node feature extraction, failing to capture the ordered relationships between API calls.

The related work section of this paper reviews various approaches to malware detection and classification, focusing on methods that utilize both traditional machine learning and deep learning algorithms. Traditional machine learning techniques have explored API sequences but often overlook the invocation relationships and structural information between APIs, leading to limited feature representation. In contrast, deep learning methods, including those using RNNs and GCNs, have shown improved accuracy by leveraging more

complex representations of API sequences. However, many of these approaches simplify API sequences into fixed-length representations, neglect ordered relationships, or primarily focus on first-order graph information, thus missing higher-order structural insights. This review highlights the strengths and limitations of these methods, emphasizing the need for more comprehensive models that can effectively capture the full scope of API sequences, including both their sequential and structural characteristics.

## Materials and methods

### Design overview

This section presents our proposed malware classification method, which leverages directed API call relationships. As shown in Fig 1, the process begins by addressing dataset imbalance through the application of the SMOTE algorithm. Next, a directed graph is constructed for each API sequence. From this directed graph, we extract multi-dimensional node attribute features and compute the first- and second-order adjacency matrices, which approximate the directed graph's adjacency matrix. These attribute features, along with the first- and second-order adjacency matrices, are input into the GCN, generating a feature embedding representation of the directed graph corresponding to the malware file's API sequence. Subsequently, this two-dimensional feature embedding is transformed into a grayscale image. Exploiting the strengths of CNNs in image classification, the grayscale image—encapsulating attribute, structural, and directional information—is then input into the CNN for classification.

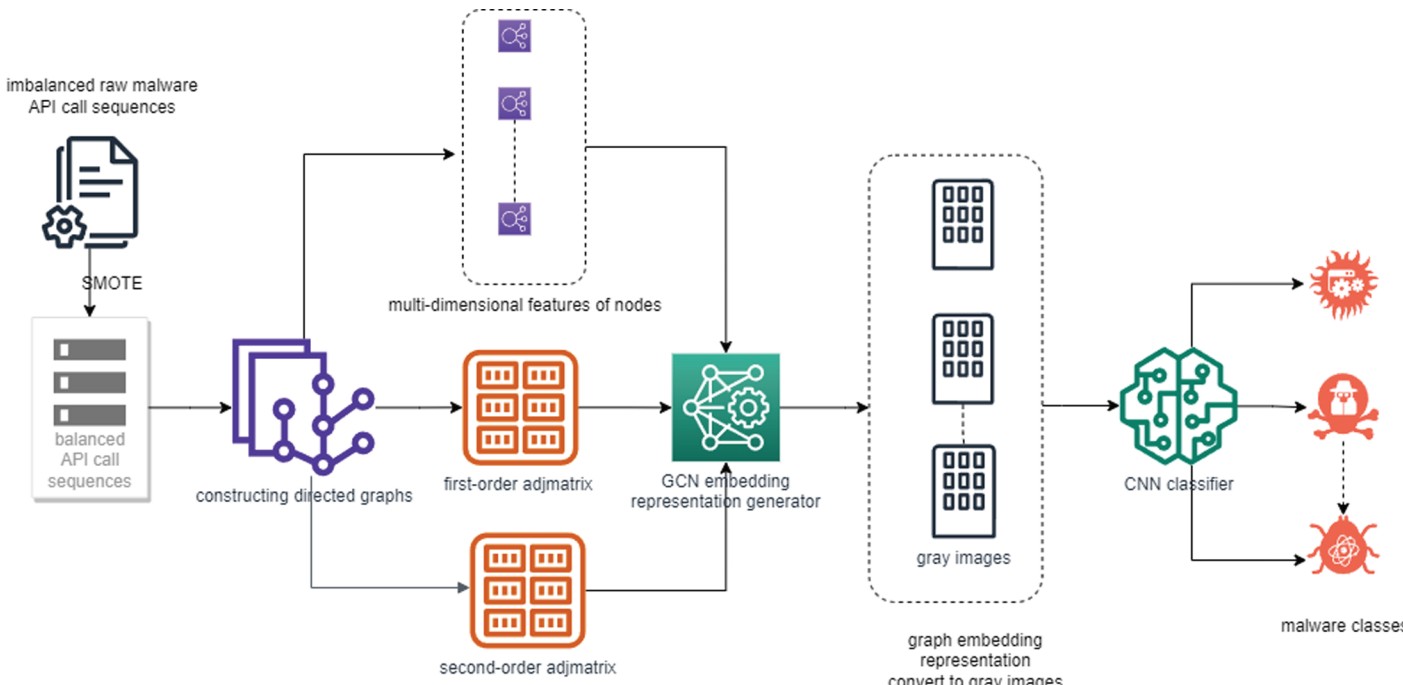

**Fig 1. Workflow of the proposed malware classification method based on FSGCN.** This figure illustrates the process from handling imbalanced datasets using the SMOTE algorithm, constructing directed graphs for API sequences, extracting multi-dimensional attribute features, computing first-order and second-order adjacency matrices, generating feature embeddings with GCN, converting these embeddings into grayscale images, and finally classifying the images using CNN.

## Notation and problem definition

For an undirected graph $G = \{X, E, A\}$, where $V = \{v_1, v_2, ..., v_N\}$ is the set of $N$ vertices of graph $G$, and $E$ is the set of edges in $G$. Here, $X \in \mathbb{R}^{N \times D}$ denotes the node feature matrix, and $A \in \mathbb{R}^{N \times N}$ represents the adjacency matrix of graph $G$. If each edge $(v_i, v_j) \in E$ has a weight $w_{i,j} > 0$, the graph $G$ is called a weighted graph. If the graph is directed, denoted as $\vec{G}$, then $\vec{A}$ represents the adjacency matrix of the directed graph $\vec{G}$, where $(v_i, v_j) \neq (v_j, v_i)$ and the edge weights $w_{i,j} \neq w_{j,i}$.

## First-order and second-order graph convolutional networks (FSGCN)

In the current landscape of deep learning methods for extracting features from malware API sequences, most approaches are grounded in NLP [29] and GCN [35]. These methods typically begin by extracting features from the malware API sequences, followed by classification. In NLP-based approaches, however, API sequences are often treated as simple text, neglecting the structural features inherent in the relationships between API calls. This oversight results in reduced performance. In contrast, GCN-based methods capture structural information within API sequences, but GCNs operate on undirected graphs, inherently discarding directional relationships. This limitation also leads to reduced performance.

To address these limitations, we propose first-order and second-order graph convolutional networks to extract feature representations from directed graphs generated by API sequences. Drawing inspiration from DGCN [36], we now explore how FSGCN effectively extracts both structural and directional information.

**Graph convolutional network.** Unlike regular data structures, graph structures are irregular and do not exhibit translational invariance. This characteristic makes traditional CNNs ill-suited for processing graph data. To overcome this limitation, Graph Convolutional Networks (GCNs) were developed. GCNs, similar to CNNs, focus on feature extraction but are specifically designed to handle graph data [37]. In this paper, we focus on multi-layer GCNs. This type of GCN follows a layered propagation rule, as shown in Eq (1):

$$H^{(l+1)} = \sigma\left(\tilde{D}^{-\frac{1}{2}} \tilde{A} \tilde{D}^{-\frac{1}{2}} H^{(l)} W^{(l)}\right), \tag{1}$$

where $\tilde{A} = A + I_N$ represents the adjacency matrix of the undirected graph $G$ with self-loops, $\tilde{D}$ is the degree matrix of $\tilde{A}$. $W^{(l)}$ is the trainable weight matrix for the $l$-th layer, $\sigma(\cdot)$ is the activation function, $H^{(l)}$ is the latent representation output by the $l$-th layer of the GCN, and $H^{(0)} = X$.

Spectral-based GCN methods are inherently designed for undirected graphs, relying on a symmetric Laplacian matrix. This characteristic makes spectral-based GCN methods incompatible with directed graphs. As a result, these methods disregard directionality, causing a loss of crucial directional information inherent to the data. This limitation hinders the accurate representation of directed graphs' structure. To address this, our approach uses first-order and second-order graph convolutions to approximate directed graph convolutions. By using these approximations, we preserve the directional features of the graph, maintaining the integrity of the original data structure.

**First-order and second-order graph convolutional network framework.** As is well known, the inputs of GCN methods include the node feature matrix $X$ and the adjacency matrix $A$ of the graph. So, what are the feature matrix and adjacency matrix in our FSGCN ? Next, we provide the definitions of the first-order and second-order adjacency matrices for FSGCN.

**Definition 1 (first-order adjacency matrix).** We define the first-order adjacency matrix $A_F$ of a directed graph $\vec{G}$ as follows:

$$A_F(i,j) = A(i,j), \tag{2}$$

where nodes $i$ and $j$ are connected by edge $(i,j) \in E$, and $A(i,j)$ is the adjacency matrix of the undirected graph $G$ corresponding to the directed graph $\vec{G}$, which is symmetric.

The resulting first-order adjacency matrix is a symmetric adjacency matrix. Unfortunately, $A_F$ omits directional information. To preserve this crucial directionality, we approximate the directional information of the directed graph using a second-order adjacency matrix. In this approach, if two vertices share a common neighbor, they are considered similar. Therefore, we define the second-order adjacency matrix based on this principle of shared neighboring vertices.

**Definition 2 (second-order adjacency matrix).** The second-order adjacency matrix is defined as the matrix of normalized weights of edges shared by neighbors. It is divided into the second-order in-degree adjacency matrix $A_{S_{in}}$ and the second-order out-degree adjacency matrix $A_{S_{out}}$,

$$A_{S_{in}}(i,j) = \sum_k \frac{\vec{A}_{k,i}\vec{A}_{k,j}}{\sum_v \vec{A}_{k,v}}, \tag{3}$$

$$A_{S_{out}}(i,j) = \sum_k \frac{\vec{A}_{i,k}\vec{A}_{j,k}}{\sum_v \vec{A}_{v,k}}, \tag{4}$$

where $k, v$ belong to the set of vertices $V$. $A_{S_{in}}(i,j)$ represents the normalized sum of the weights of all edges from vertex $v_k$ to vertices $v_i$ and $v_j$, which is the weighted in-degree of vertices $v_i$ and $v_j$ with common neighbors. $A_{S_{out}}(i,j)$ represents the normalized sum of the weights of all edges from vertices $v_i$ and $v_j$ to vertex $v_k$, which is the weighted out-degree of vertices $v_i$ and $v_j$ with common neighbors.

To clarify Definitions 1 and 2, we provide a detailed example. In Fig 2(A), the first-order neighbors of vertex 1 are vertices 6, 3, and 4, with the directed graph treated as undirected. In Fig 2(B), vertices 2 and 5 point to vertices 6 and 3. Thus, both $A_{S_{in}}(3,6)$ and $A_{S_{in}}(6,3)$ are the sum of the normalized weights of edges (2,3), (2,6), (5,3), and (5,6), corresponding to the second-order in-degree adjacency matrix value between vertices 3 and 6. Similarly, in Fig 2(C), vertices 3 and 4 point to vertices 7 and 8, indicating that vertices 7 and 8 are common neighbors of both. The second-order out-degree adjacency matrix values, $A_{S_{out}}(3,4)$ and $A_{S_{out}}(4,3)$, are the sum of the normalized weights of edges (3,7), (4,7), (3,8), and (4,8). Thus, $A_{S_{in}}$ and $A_{S_{out}}$ are symmetric matrices.

Using the definitions of the first-order and second-order adjacency matrices, we obtain the first-order and second-order adjacency matrices, $A_F$, $A_{S_{in}}$ and $A_{S_{out}}$, for the directed graph. These three symmetric adjacency matrices approximate the adjacency matrix $\vec{A}$ of the directed graph, aiding subsequent graph convolution operations.

In GCN operations, in addition to constructing the adjacency matrix, the node feature attributes of the graph are also required. However, the malware dataset does not have specified API type attributes. To fully utilize the power of GCN, we need to construct some API-related feature attributes. Based on this motivation, we designed the node attributes shown in Table 1, which can be computed using the following formulas:

$$api\_num = Count(\vec{G}), \tag{5}$$

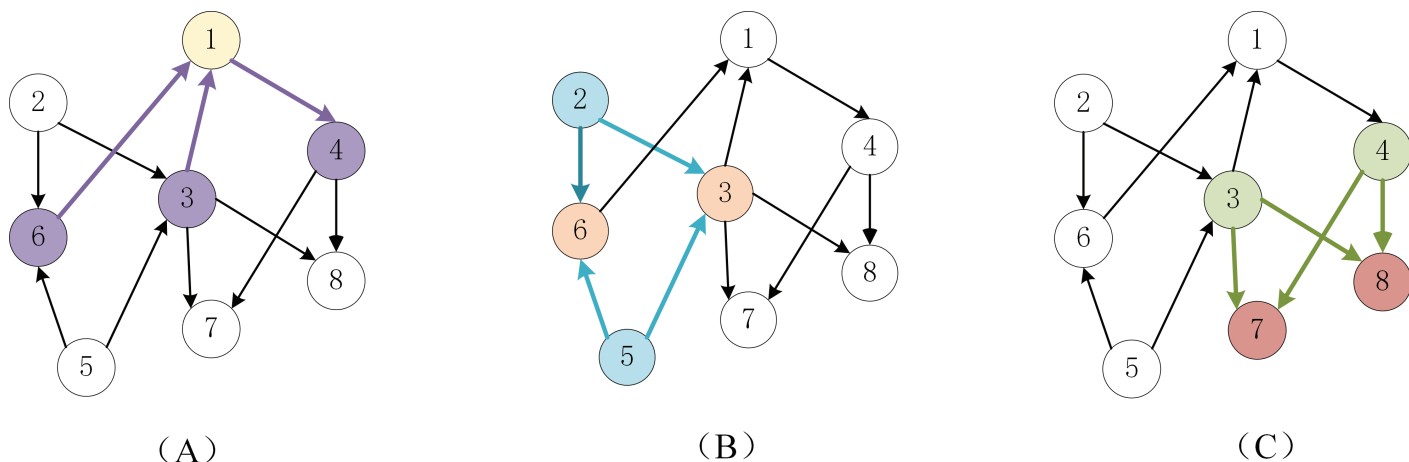

**Fig 2. An example of computing the first-order and second-order adjacency matrices for a graph.** A:Example of first-order vertex computation. B: Example of Second-order-indegree . C: Example of Second-order-outdegree.

**Table 1. Descriptions of the API node attributes.**

| Attribute Name | Attribute Description |
| --- | --- |
| *api_num* | The amount number of each API |
| *api_in* | The input degree of each API |
| *api_out* | The output degree of each API |
| *api_in_sum* | The sum of in-degree weight of each API |
| *api_out_sum* | The sum of out-degree weight of each API |
| *api_eigenvector_centrality* | The eigenvector centrality of each API |
| *api_pagerank* | The PageRank value of each API |

$$api\_in\,(v_i) = \sum_{j=1}^{N} \vec{A}_{ji}, \tag{6}$$

$$api\_out\,(v_i) = \sum_{j=1}^{N} \vec{A}_{ij}, \tag{7}$$

$$api\_in\_sum\,(v_i) = \sum_{j=1}^{N} w_{ji}, \tag{8}$$

$$api\_out\_sum\,(v_i) = \sum_{j=1}^{N} w_{ij}, \tag{9}$$

where $Count\,(\cdot)$ calculates the number of nodes in the graph, and $\vec{A}_{ji}$ is the element in the $j$-th row and $i$-th column of the directed adjacency matrix $\vec{A}$, indicating whether there is an edge from vertex $v_j$ to vertex $v_i$. If such an edge exists, then $\vec{A}_{ji} = 1$; otherwise, $\vec{A}_{ji} = 0$. Similarly, $\vec{A}_{ij}$ is the element in the $i$-th row and $j$-th column of the directed adjacency matrix $\vec{A}$, indicating whether there is an edge from vertex $v_i$ to vertex $v_j$. If such an edge exists, then $\vec{A}_{ij} = 1$; otherwise, $\vec{A}_{ij} = 0$. $w_{ji}$ represents the edge weight from vertex $v_j$ to vertex $v_i$, and $w_{ij}$ represents the edge weight from vertex $v_i$ to vertex $v_j$.

In addition, to capture the more complex relationships within the API sequences, we also designed the eigenvector centrality and PageRank values for the vertices of the API sequence directed graph. Eigenvector centrality not only considers the degree of the vertex itself but also takes into account the centrality of its neighbors; in other words, the importance of other

vertices connected to a given vertex also influences its centrality. Specifically, the eigenvector centrality $C_{api}(v_i)$ of vertex $v_i$ is defined by the following relation:

$$C_{api}(v_i) = \frac{1}{\lambda} \sum_{j=1}^{N} \vec{A}_{ij} C_{api}(v_j),$$

(10)

where $C_{api}(v_i)$ is the eigenvector centrality of vertex $v_i$, and $\lambda$ is the eigenvalue corresponding to the eigenvector $C_{api}(v_i)$. Here, $C_{api}$ corresponds to the api_eigenvector_centrality in Table 1.

The core of the PageRank algorithm is based on the concept of assessing the importance of web pages using the link structure of a network graph. This metric is not only essential for search engine optimization but also highly valuable in graph classification. In graph classification, the PageRank value of a node can be utilized as a feature, in combination with other structural features such as node degree and neighborhood information. By incorporating the PageRank value, the model is better equipped to discern the relative importance of each node within the entire graph, thereby enriching the classification process with more nuanced information. The formula for calculating the PageRank value of a node is as follows:

$$PageRank(v_i) = \frac{1-d}{N} + d \sum_{v_j \in M(v_i)} \frac{PageRank(v_j)}{L(v_j)},$$

(11)

where $PageRank(v_i)$ is the PageRank value of the node, and $d$ is the damping factor, typically set around 0.85. It represents the probability that a random walker will continue jumping to other nodes according to the network's link structure, while $1-d$ represents the probability of randomly jumping to any node, The term $M(v_i)$ is the set of all nodes that point to node $v_i$, and $L(v_i)$ is the total number of nodes that node $v_i$ points to (i.e., the out-degree). Here, the *PageRank* value corresponds to api_pagerank in Table 1.

Through the above calculations, we obtain the feature matrix $\vec{X} = [api\_num, api\_in, api\_out, api\_in\_sum, api\_out\_sum, C_{api}, PageRank]$ for the directed graph of the API sequence. Therefore, based on the concept of GCN, we can derive the first-order graph embedding representation $Z_F$, the second-order in-degree graph embedding representation $Z_{S_{in}}$, and the second-order out-degree graph embedding representation $Z_{S_{out}}$. The detailed calculations are as follows:

$$Z_F = \tilde{D}_F^{-\frac{1}{2}} \tilde{A}_F \tilde{D}_F^{-\frac{1}{2}} \vec{X} W,$$

(12)

$$Z_{S_{in}} = \tilde{D}_{S_{in}}^{-\frac{1}{2}} \tilde{A}_{S_{in}} \tilde{D}_{S_{in}}^{-\frac{1}{2}} \vec{X} W,$$

(13)

$$Z_{S_{out}} = \tilde{D}_{S_{out}}^{-\frac{1}{2}} \tilde{A}_{S_{out}} \tilde{D}_{S_{out}}^{-\frac{1}{2}} \vec{X} W,$$

(14)

here, $\tilde{A}_F$ is the normalization of $A_F$, $\tilde{A}_{S_{in}}$ is the normalization of $A_{S_{in}}$, and $\tilde{A}_{S_{out}}$ is the normalization of $A_{S_{out}}$. $\tilde{D}_F = diag\left(\sum_j^N \tilde{A}_F(i,j)\right)$, $\tilde{D}_{S_{in}} = diag\left(\sum_j^N \tilde{A}_{S_{in}}(i,j)\right)$ and $\tilde{D}_{S_{out}} = diag\left(\sum_j^N \tilde{A}_{S_{out}}(i,j)\right)$.

Subsequently, we combine $Z_F$, $Z_{S_{in}}$ and $Z_{S_{in}}$ through a concatenation function to obtain an embedding representation $Z$ that incorporates feature attribute information, structural information, and directional information. The calculation is as follows:

$$Z = Concat\left(Z_F, \alpha Z_{S_{in}}, \beta Z_{S_{out}}\right),$$

(15)

here, $\alpha$ and $\beta$ are balancing parameters used to adjust the importance of first-order and second-order graph information. Finally, the final feature representation $\hat{Z}$ is obtained through a nonlinear activation function, as follows:

$$\hat{Z} = ReLU(Z). \tag{16}$$

The overall process of the first-order and second-order graph convolutional networks is shown in Fig 3.

**Malware classification based on FSGCN.** Most malware datasets suffer from imbalanced data distributions, as shown in Fig 4. This imbalance can cause models to favor the majority class during training, leading to reduced overall classification performance. To mitigate this, we implemented sample resampling techniques, which are commonly used to handle class imbalance. The two main strategies are oversampling and undersampling [38]. In Fig 4(A), an undersampling approach would remove samples from the majority class. However, this may reduce the amount of training data and cause the loss of valuable features, negatively affecting model performance. Therefore, we chose to employ oversampling techniques to better balance the class distribution.

Oversampling increases the representation of the minority class by generating more samples. Among the various methods, Random Oversampling and SMOTE (Synthetic Minority Over-sampling Technique) [39] are the most widely used. Random Oversampling duplicates minority class samples and adds them to the training set, effectively balancing the sample sizes. Though simple, this method is highly effective.

On the other hand, SMOTE generates synthetic samples by identifying nearest neighbors in the feature space of the minority class and interpolating between them. This not only

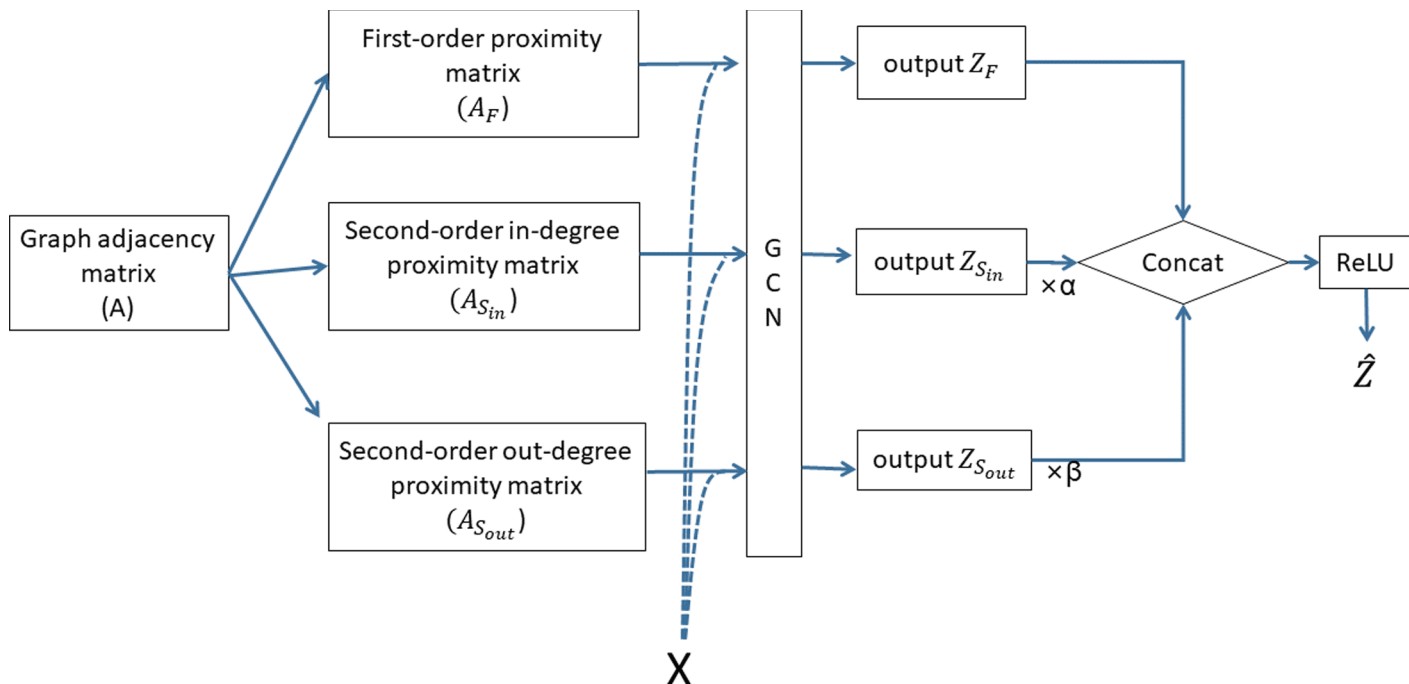

**Fig 3. The overall process of the first-order and second-order graph convolutional networks.**

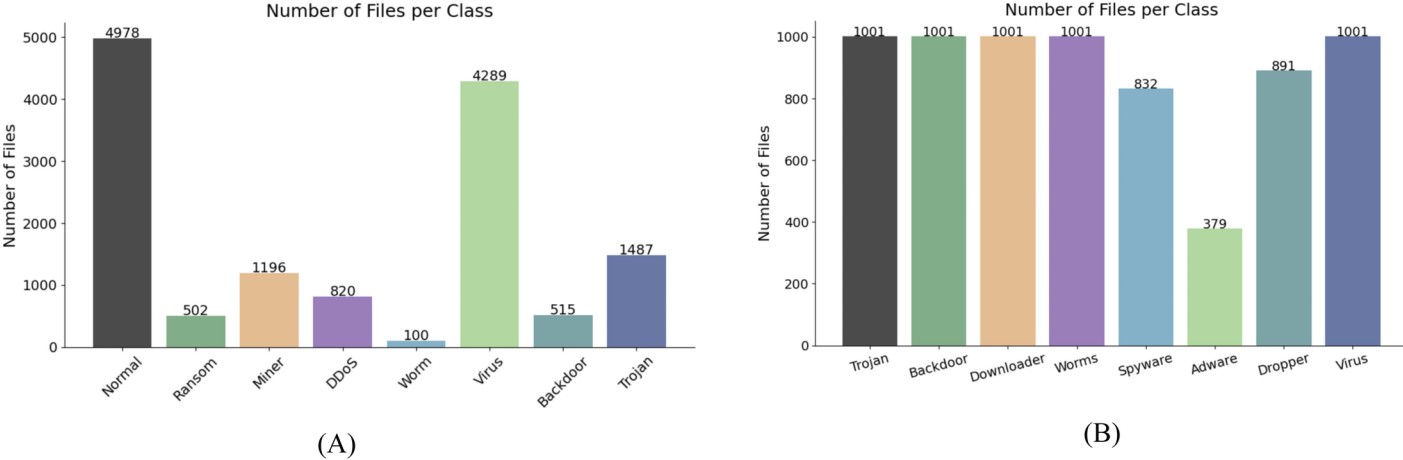

**Fig 4. The distribution of samples in Dataset 1 and Dataset 2. (A) is Dataset1, (B) is Dataset2.**

increases the number of minority class samples but also diversifies the sample pool, avoiding the issues of simple duplication. Given its ability to enhance sample diversity, we chose to apply the SMOTE algorithm for dataset resampling.

A CNN is a deep learning model that excels at processing grid-like data, such as images. Its convolutional layers automatically learn local features through convolution operations. In our approach, the malware API sequence is processed by the FSGCN model, producing a directed graph feature representation, $\hat{Z}$. This two-dimensional feature matrix encapsulates the graph's attribute features, structural, and directional information. When converted into a grayscale image, a CNN can be used for image classification. The calculation for converting $\hat{Z}$ into a grayscale image is as follows:

$$Z_{gray}(i,j) = \left\lfloor \frac{\hat{Z}(i,j) - a}{b - a} \times 255 \right\rfloor, \tag{17}$$

where $Z_{gray}(i,j)$ is the value at position $(i,j)$ in the converted grayscale image, $a$ is the minimum value in $\hat{Z}$, $b$ is the maximum value in $\hat{Z}$. The notation $\lfloor \cdot \rfloor$ represents the floor function, ensuring that the pixel value is an integer.

Next, we design a two-layer CNN classifier, where each layer uses a convolutional kernel of size $k = 3 \times 3$, with a stride of $s = 1$ and padding of $p = 1$. After each convolutional layer, a max-pooling layer is applied, followed by three fully connected layers to compute the final classification result. For the CNN classifier, we use the cross-entropy loss function to quantify the classification loss, as shown in the following formula:

$$L_C(p,q) = -\sum_{i=1}^{C} p_i \log(q_i), \tag{18}$$

where $C$ represents the number of classes, $p_i$ represents the true value, and $q_i$ represents the predicted value.

The detailed process of the FSGCN-based malware classification algorithm is shown in Algorithm 1.

**Algorithm 1** Malware classification based on FSGCN algorithm with algpseudocode

1: **procedure** Our Algorithm Procedure
2: **Input:** Dataset; The number of malicious file categories $K$; The ID of each malicious file $Fid$; The ID of each malicious file $L(K)$; Maximum iterations $T$.
3: **Output:** The classification results.
4: Use the SMOTE algorithm to handle imbalanced datasets.
5: **for** i=1 to $K$ **do**
6: **for** Fid in $L(K)$ **do**
7: Obtain the API sequence for the file with ID $Fid$, denoted as $Seq$.
8: Use Seq to construct the directed graph $\vec{G}$ and its corresponding undirected graph $G$.
9: Obtain the adjacency matrix $A$ of $G$ and assign it to $A_F$ using Eq (2).
10: Obtain the adjacency matrix $\vec{A}$ from the directed graph $\vec{G}$, and calculate $A_{S_{in}}$ and $A_{S_{out}}$ using Eqs (3) and (4).
11: Calculate the node attribute '*api_num*', '*api_in*', '*api_out*', '*api_in_sum*' and '*api_out_sum*' from the directed graph $\vec{G}$ using Eqs (5), (6), (7), (8), and (9).
12: Obtain the values of '*api_eigenvector_centrality*' and '*api_pagerank*' using Eqs (10) and (11), respectively.
13: Construct the node attributes $X$ of the directed graph $\vec{G}$ based on the values from Eqs (5), (6), (7), (8), (9), (10), and (11).
14: Feed $A_F$ and $X$ into Eq (12) to obtain the first-order GCN output $Z_F$.
15: Feed $A_{S_{in}}$ and $X$ into Eq (13) to obtain the second-order in-degree GCN output $Z_{S_{in}}$.
16: Feed $A_{S_{out}}$ and $X$ into Eq (14) to obtain the second-order out-degree GCN output $Z_{S_{out}}$.
17: Subsequently, calculate using Eqs (15) and (16) to obtain the latent representation $\hat{Z}$ of the directed graph for the malware file's API sequence.
18: Save it in the latent representation list $L_{\hat{z}}$.
19: **end for**
20: **end for**
21: Convert the two-dimensional latent representation list $L_{\hat{z}}$ into grayscale images using Eq (17).
22: Divide these grayscale images into a training set (train), a test set (test), and a validation set (val) in a 7:1.5:1.5 ratio.
23: **for** j=1 to $T$ **do**
24: Feed the train and test datasets into the CNN classifier to obtain the classification results.
25: **end for**
26: **return** The classification results.
27: **end procedure**

## Results and discussion

To verify the effectiveness of our model, we conducted evaluations on two real-world datasets. These tests involved a comparative analysis of our model's results against those achieved by several prevalent malware classification methods, as well as a comparison with findings from related research studies.

## Datasets and experimental environment

This study utilized two distinct datasets. **Dataset 1**, sourced from the Aliyun Tianchi competition (Aliyun Security Malware Detection), consists of API sequence samples from Windows executable files simulated in sandbox environments following internet execution [40]. These files, all Windows binaries, have been desensitized for security reasons. The dataset contains various types of malicious software, including viruses, Trojans, mining programs, DDoS Trojans, and ransomware, totaling 13,887 files. The data is divided into eight categories: normal, ransomware, mining programs, Trojans, worm viruses, infected viruses, backdoors, and Trojans. Notably, this dataset does not capture ordering relationships between thread IDs (TIDs); instead, the index within each TID indicates the calling order. Since software often runs multiple threads, we simplify this by considering the index order within a file as the calling order. The distribution of Dataset 1 is shown in Fig 4(A).

**Dataset 2** is a publicly available malware dataset created using Cuckoo Sandbox, based on Windows OS API call analysis [41]. This dataset is notable for being the first to include metamorphic malware in sequential API call datasets. It categorizes malware into eight families: Trojan, Backdoor, Downloader, Worm, Spyware, Adware, Dropper, and Virus, with 7,107 malware instances. The distribution of Dataset 2 is shown in Fig 4(B). A summary of both datasets is provided in Table 2.

In our experiments, data was randomly sampled, with 70% allocated to training and 15% each to validation and testing sets.

The model was executed and tested on a system with an 11th Gen Intel(R) Core(TM) i5-11600KF @ 3.90 GHz CPU, 32 GB RAM, and a GeForce RTX 3060 Ti GPU, running Windows 10.

## Evaluation metrics

In this study, we evaluate the performance of our model using four key metrics: accuracy, recall, precision, and F1 score. Let $TP$ (True Positive) represent the number of samples correctly classified by the model as belonging to a particular malware category. $TN$ (True Negative) refers to the number of samples correctly classified as not belonging to a specific malware category. $FP$ (False Positive) indicates the number of instances where the model incorrectly classifies samples that do not belong to a particular malware category as being part of it. $FN$ (False Negative) refers to the number of cases where the model incorrectly classifies samples that belong to a specific malware category as not being part of it.

Accordingly, accuracy is defined as $\frac{TP+TN}{TP+FN+TN+FP}$, representing the proportion of all samples correctly classified by the model across all categories. Precision, calculated as $\frac{TP}{TP+FP}$, indicates

**Table 2. Dataset overview.**

| Dataset | Source | Total Samples | Categories | Category Distribution | Data Split (Train/Validation/Test) |
|---|---|---|---|---|---|
| Dataset 1 | Aliyun Tianchi Competition (Aliyun Security Malware Detection) | 13887 | 8 | Normal, Ransomware, Mining Programs, Trojans, Worm Viruses, Infected Viruses, Backdoor Programs, Trojan Programs | 70%/15%/15% |
| Dataset 2 | Cuckoo Sandbox (Publicly Available) | 7107 | 8 | Trojan, Backdoor, Downloader, Worms, Spyware Adware, Dropper, Virus | 70%/15%/15% |

the proportion of samples correctly classified as belonging to a specific malware category out of all the samples the model classifies as belonging to that category. Recall, defined as $\frac{TP}{TP+FN}$, measures the proportion of samples correctly classified by the model as belonging to a specific malware category out of all actual samples in that category. The F1 Score, computed as $2 * \frac{Precision*Recall}{Precision+Recall}$, is the harmonic mean of precision and recall, providing a balanced measure of the model's accuracy and completeness.

## Experimental results

Before conducting our experiments, we first established the hyperparameters for our model. For Dataset 1, we set the values of $\alpha$ and $\beta$ in the FSGCN algorithm (Eq 15) to 1, and configured the GCN to have one layer. In the CNN configuration, the training, validation, and testing datasets were allocated in a 7:1.5:1.5 ratio, with a batch size of 100. The learning rate was set to 0.001. The CNN architecture consisted of one convolutional layer, one fully connected layer, and used cross-entropy loss as the loss function.

For Dataset 2, we kept the same hyperparameters for the FSGCN algorithm as in Dataset 1. In the CNN setup for Dataset 2, we adjusted the learning rate to 0.0001 and introduced a regularization term with a weight decay of 0.1, while keeping the other settings consistent with those in Dataset 1.

**Graph embedding representations transformed into grayscale images.** After computing the graph embedding representations for each sample, as shown in Eq (17), we observed variability in their size, attributed to the differing number of graph nodes per sample. Upon further analysis, we identified 295 API call types in Dataset 1 and 278 in Dataset 2. Leveraging the proficiency of CNNs in image recognition and classification, we transformed the graph embeddings into uniform two-dimensional forms and normalized their values to fall within the 0-255 range.

This transformation resulted in the graph embeddings for Dataset 1 having dimensions of $(295, 3 \times D)$, where $D$ is the length of the feature vector column $X$, and $3 \times D$ is derived using Eq (15). Similarly, for Dataset 2, the dimensions were $(278, 3 \times D)$. We then visualized the transformed data as normalized grayscale images. For instance, Fig 5 displays the grayscale images of three distinct software types from Dataset 1: Normal, Ransom, and Miner, as shown in Fig 5A, 5D, and 5G, respectively. A notable observation is the significant differences in the grayscale representations of the graph embeddings for these software types, while images within the same category (e.g., Fig 5A, 5B, and 5C) exhibit strikingly similar grayscale patterns. This consistency underscores the effectiveness of our proposed multi-dimensional directed graph embedding algorithm in extracting distinctive features from various software types.

**Comparison with baselines.** To demonstrate the effectiveness of our methodology, we conducted experiments on the two previously discussed datasets using our algorithm. Additionally, we conducted a comparative analysis with two categories of malware classification algorithms: traditional and graph-based methods. The specifics of these experiments are as follows:

For traditional malware classification algorithms, we selected four well-established methods: Multilayer Perceptron (**MLP**) [42],**N-gram** [43], **LSTM** [44], and **SVM** [45]. The MLP, a deep learning-based method, uses node features for classification. Our implementation consists of a three-layer MLP. The input and output dimensions are determined by the number of APIs and the types of malware, respectively. N-gram and LSTM are used as feature-generation methods, with N-gram examining gram sizes of 1, 2, and 3, while LSTM employs two hidden

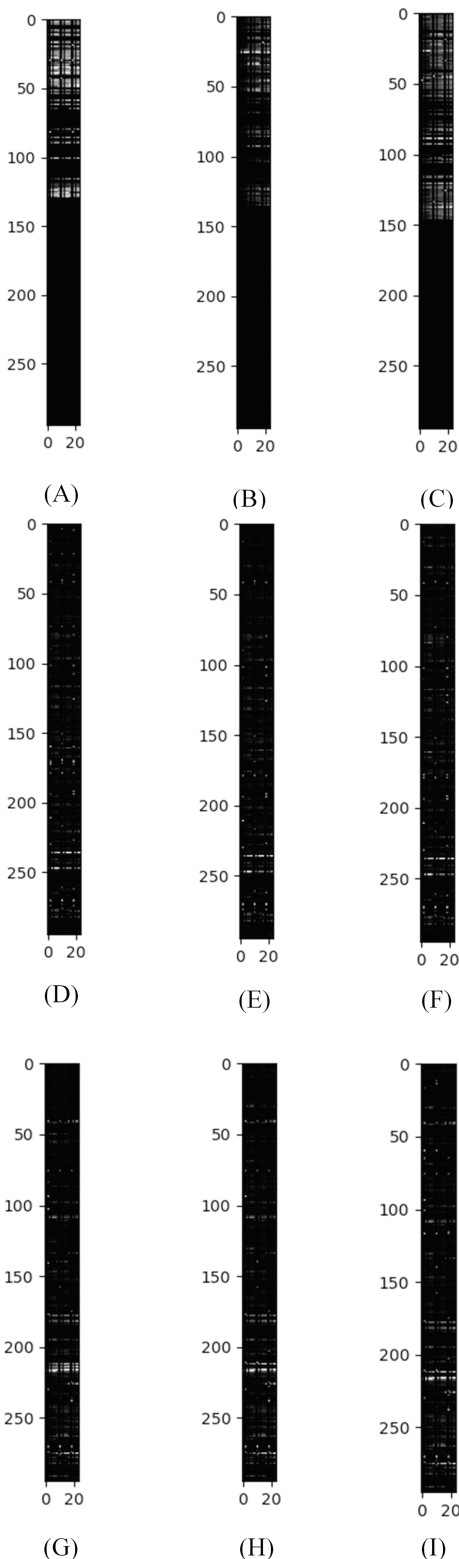

**Fig 5. Examples of normalized grayscale images for samples of three different types of malware graph embedding representations after transformation in Dataset 1.** (a), (b), and (c) belong to the Normal type, while (d), (e), and (f) belong to the Ransom type, and (g), (h), and (i) belong to the Miner type.

layers, each with 1000 units. The SVM algorithm, which uses a linear kernel and a one-vs-rest strategy, is also included in our comparison.

In the realm of graph-based classification algorithms, we used three prominent graph neural network algorithms: **GCN** [37], Graph Sample and Aggregated Graphsage Embedding (**GraphSAGE**) [46], and Graph Attention Network (**GAT**) [47]. GCN, an unsupervised learning method, excels at handling graph data, extracting node features, and understanding the topological structures between nodes. GraphSAGE, another unsupervised technique, generates node embeddings using neighboring information and demonstrates proficiency in node classification and community detection. GAT employs an attention mechanism to dynamically learn the weight relationships between nodes and their neighbors, offering robust representation capabilities and scalability.

The comparative results of our approach and the four traditional malware classification methods are shown in Table 3. Our method significantly surpasses the others in accuracy, achieving scores of 0.999 and 0.676 on Dataset 1 and Dataset 2, respectively. Furthermore, our approach outperforms the other four methods in Recall, Precision, and F1-score. These results demonstrate that our proposed method outperforms the MLP, N-gram, LSTM, and SVM algorithms in malware classification.

Table 4 presents a comparison of the classification performance between our method and classical graph embedding algorithms—GCN, GraphSAGE, and GAT—specifically focusing on Dataset 1. The table outlines the performance metrics for our method and the three graph-based malware classification algorithms—GCN, GraphSAGE, and GAT—across four key metrics: Accuracy, Recall, Precision, and F1 Score. The data reveals that both our algorithm and the GraphSAGE algorithm outperform the other two. However, overall, our method slightly outperforms the GraphSAGE algorithm in malware classification.

Fig 6(a)–6(d) presents the loss curves for GCN, GraphSAGE, GAT, and our model, showing their performance on both the training and validation sets. In our experiments, GCN, GraphSAGE, GAT, and our model were configured with identical learning rates and optimizers to ensure consistency in the comparison. To provide a clearer view of model convergence, training was extended to 200 epochs. As shown in Fig 6(a)–6(d), our model exhibits a notably rapid decrease in its loss curve during the training phase, demonstrating its efficient learning capability. Regarding convergence stability, our model maintains a relatively low loss value at convergence, with less variability compared to the other three models. Moreover, while

**Table 3. Performance compared with traditional malware classification methods on two datasets.**

| Algorithms | Dataset1 [40] | | | | Dataset2 [41] | | | |
|---|---|---|---|---|---|---|---|---|
| | *Accuracy* | *Recall* | *Precision* | *F1* | *Accuracy* | *Recall* | *Precision* | *F1* |
| MLP [42] | 0.871 | 0.733 | 0.846 | 0.781 | 0.572 | 0.572 | 0.573 | 0.571 |
| N-gram [43] | 0.936 | 0.910 | 0.865 | 0.887 | 0.616 | 0.616 | 0.620 | 0.617 |
| LSTM [44] | 0.942 | 0.893 | 0.885 | 0.889 | 0.476 | 0.497 | 0.491 | 0.482 |
| SVM [45] | 0.839 | 0.839 | 0.834 | 0.835 | 0.553 | 0.562 | 0.579 | 0.564 |
| **Ours** | **0.999** | **0.999** | **0.999** | **0.999** | **0.676** | **0.676** | **0.713** | **0.675** |

**Table 4. Performance compared with graph convolution based malware classification algorithms.**

| Algorithms | *Accuracy* | *Recall* | *Precision* | *F1* |
|---|---|---|---|---|
| GCN [37] | 0.971 | 0.971 | 0.972 | 0.971 |
| GraphSAGE [46] | 0.993 | 0.993 | 0.993 | 0.993 |
| GAT [47] | 0.989 | 0.989 | 0.990 | 0.989 |
| **Ours** | **0.999** | **0.999** | **0.999** | **0.999** |

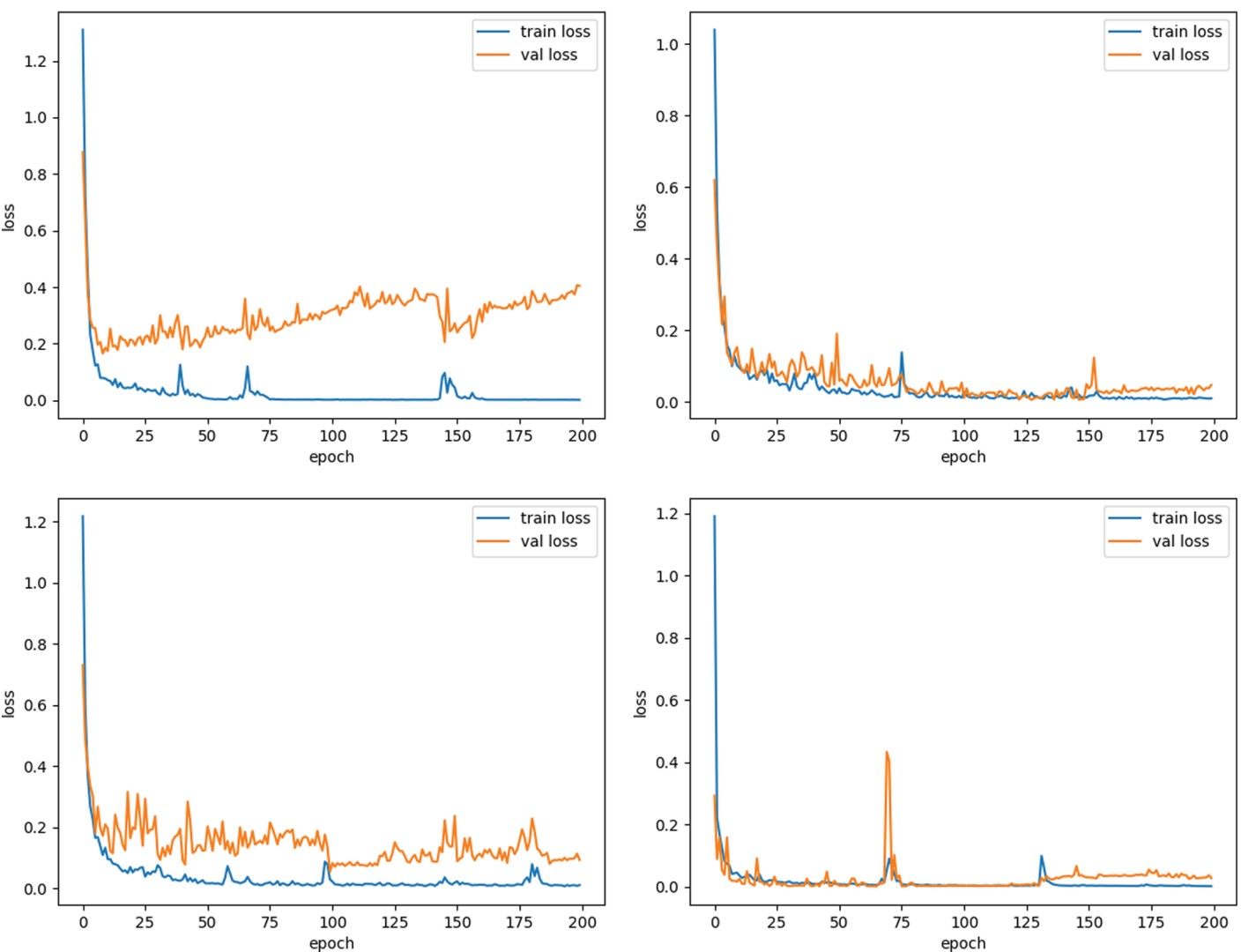

**Fig 6. The loss curves of GCN, GraphSAGE, GAT, and our model.** (a): The loss curves of GCN. (b): The loss curves of GraphSAGE. (c):he loss curves of GAT. (d):The loss curves of Our work.

the loss value on the validation set of our model is low, it is slightly higher than that of the training set, indicating a well-balanced generalization.

Table 5 presents a comparative analysis of our method against three benchmark methods in terms of malware classification performance on Dataset 1. Notably, SDGNet [30], Mal-ASSF [48], and LGMal [49] utilized Dataset 1, similar to our study,while Catak et al. [50] used Dataset 2, which corresponds to our second dataset. The results in Table 5 clearly show that our approach outperforms the other methods across key metrics such as accuracy, precision, recall, and F1 score.

**Ablation studies.** To thoroughly evaluate the effectiveness of our module, we conducted an ablation study with two distinct scenarios to assess their impact on model performance. The first scenario focused on evaluating the impact of varying node feature dimensionalities— specifically, 1-dimensional, 3-dimensional, 5-dimensional, and 7-dimensional—on the overall performance of the model. The second scenario centered on assessing the impact of using the

**Table 5. Performance Comparison of our work with state-of-arts-methods.**

| Algorithms | Accuracy | Recall | Precision | F1 | datasets |
|---|---|---|---|---|---|
| SDGNet [30] | 0.973 | 0.982 | 0.987 | 0.985 | dataset1 |
| Mal-ASSF [48] | 0.945 | 0.942 | 0.940 | 0.940 | |
| LGMal [49] | - | 0.881 | 0.878 | 0.878 | |
| **Ours** | **0.999** | **0.999** | **0.999** | **0.999** | |
| Catak et al. [50] | - | 0.470 | 0.500 | 0.470 | dataset2 |
| **Ours** | **0.676** | **0.676** | **0.713** | **0.675** | |

sample balancing algorithm SMOTE and addressing the challenges posed by imbalanced samples on the model's performance. The specifics of these experimental scenarios are detailed as follows:

Table 6 presents the classification performance of our method using node features with varying dimensionalities—1-dimensional, 3-dimensional, 5-dimensional, and 7-dimensional—across two datasets. In Dataset 1, consistently high classification results are observed across all dimensions, indicating that the directed graph structure of API sequences sufficiently represents the software features in this dataset, making additional node features less critical. Conversely, in Dataset 2, the classification performance exhibits greater variation across different feature dimensionalities. Notably, the 5-dimensional features achieve superior accuracy, recall, and F1 scores, while the 3-dimensional features excel in precision.

To further demonstrate the effectiveness of the selected five-dimensional features, we compared the precision-recall (PR) curves for four feature dimensionalities: 1, 3, 5, and 7. The graphs illustrate the performance of our detection model across various malware categories. As shown in Fig 7, our model performs best with five-dimensional features, showing improved detection performance for the malicious categories of Downloader, Worms, Spyware, Adware, Dropper, and Virus. Additionally, as depicted in Fig 4, despite the malware categories Spyware, Adware, and Dropper having fewer data samples in Dataset 2 compared to other classes, our model maintains superior classification performance for these categories. This indicates that our model does not favor the majority class in an imbalanced dataset, demonstrating robust performance in handling class imbalance.

Furthermore, we illustrate the performance of our model's node features using 1, 3, 5, and 7-dimensional features through t-SNE feature space visualization. As observed in Fig 8, when employing 3-dimensional and 5-dimensional features, our model effectively clusters similar data points into distinct groups, while 1-dimensional and 7-dimensional features

**Table 6. Ablation analysis of the dimensionality of node attribute features in our method across two datasets.**

| Feature number | Dataset1 [40] | | | | Dataset2 [41] | | | |
|---|---|---|---|---|---|---|---|---|
| | Accuracy | Recall | Precision | F1 | Accuracy | Recall | Precision | F1 |
| 1-dimensional feature | 0.999 | 0.999 | 0.999 | 0.999 | 0.597 | 0.597 | 0.640 | 0.595 |
| 3-dimensional feature | 0.999 | 0.999 | 0.999 | 0.999 | 0.670 | 0.670 | 0.730 | 0.672 |
| 5-dimensional feature | 0.999 | 0.999 | 0.999 | 0.999 | 0.676 | 0.676 | 0.713 | 0.675 |
| 7-dimensional feature | 0.999 | 0.999 | 0.999 | 0.999 | 0.656 | 0.656 | 0.670 | 0.656 |

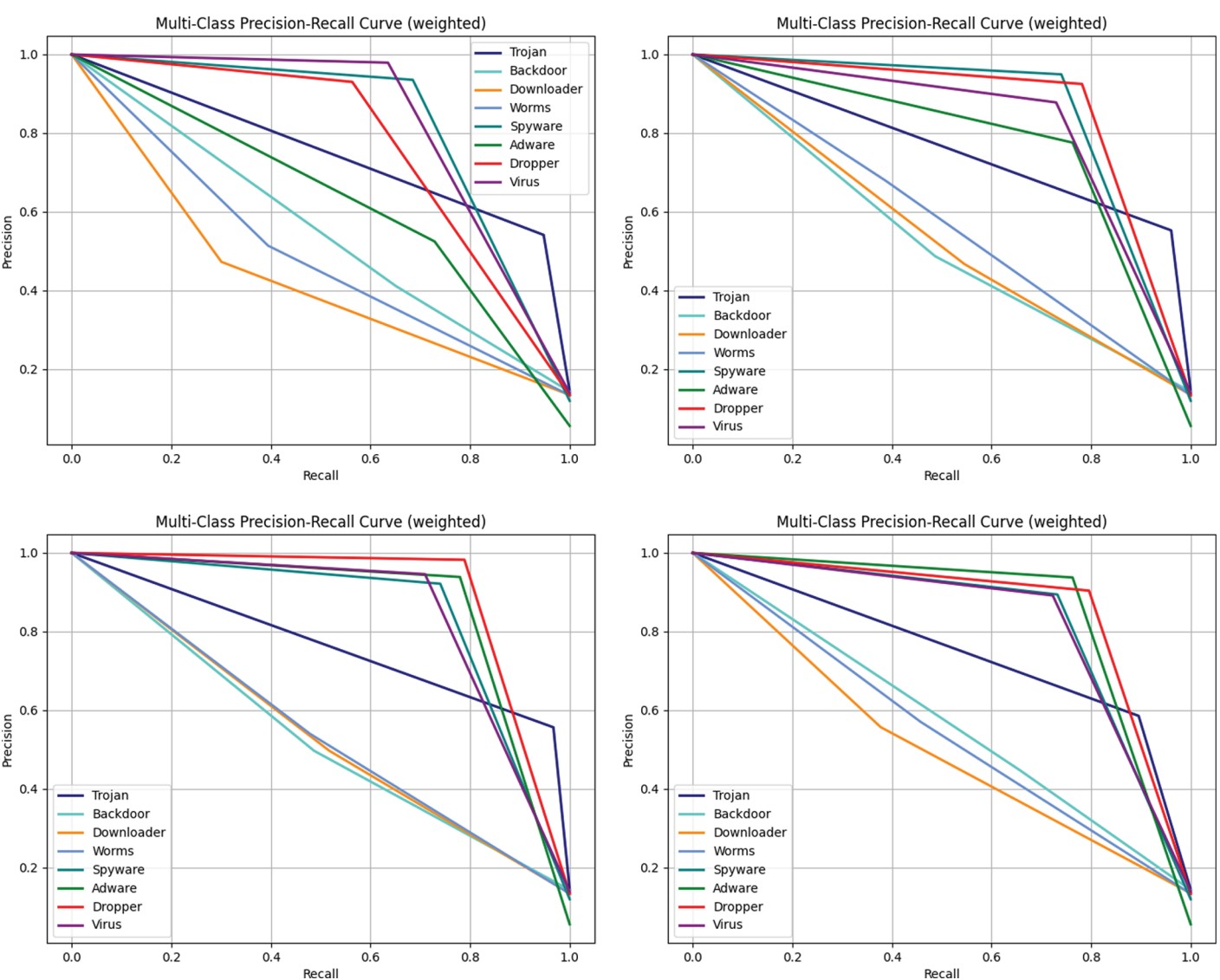

**Fig 7. Multi-dimensinal feature Precision-Recall (PR) performance of our work in dataset2.** (a): The Precision-Recall curves of 1-dimensional feature. (b): The Precision-Recall curves of 3-dimensional feature. (c):The Precision-Recall curves of 5-dimensional feature. (d):The Precision-Recall curves of 7-dimensional feature.

result in a more dispersed distribution. However, no significant distinction between the 3-dimensional and 5-dimensional features is discernible in the t-SNE visualization plots. Overall, Fig 8 reveals that by integrating multi-dimensional feature attributes, our model enhances the similarity of samples within the same category, which benefits subsequent classification tasks.

The confusion matrix in Fig 9 illustrates our model's performance in classifying eight distinct malware types. Notably, for detecting Trojan malware, the highest accuracy is achieved with 5-dimensional features, while 1-dimensional features are most effective for detecting Backdoor malware. For Downloader and Worm malware, 5-dimensional and 1-dimensional features, respectively, yield the highest detection accuracy. For Spyware, 3-dimensional features perform the best, with 5-dimensional and 7-dimensional features yielding similar

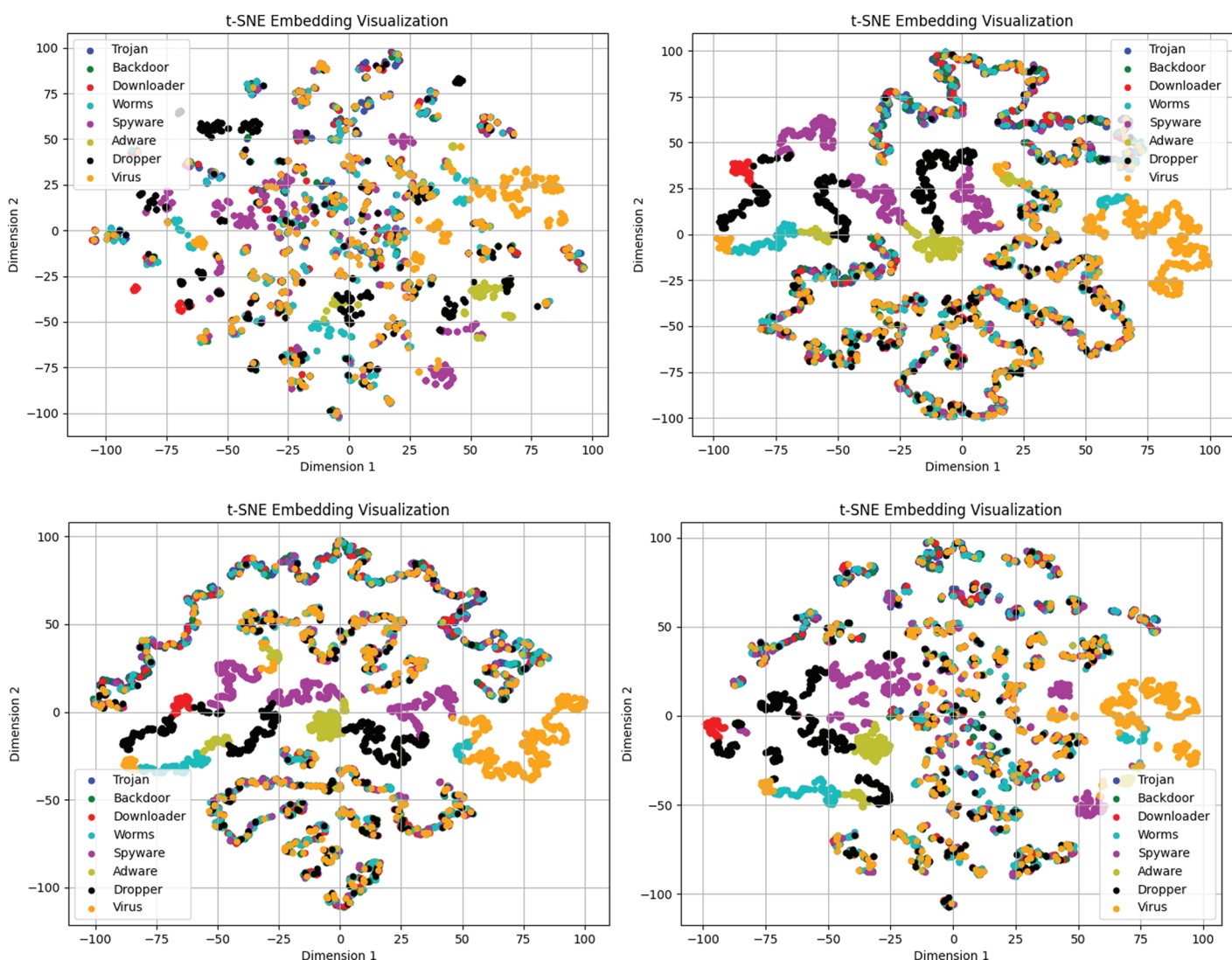

**Fig 8. Visualization of the t-SNE feature space for dimensions 1, 3, 5, and 7 of our model in dataset2.** (a): The t-SNE view of 1-dimensional feature. (b): The t-SNE view of 3-dimensional feature. (c):Thet-SNE view of 5-dimensional feature. (d):The t-SNE view of 7-dimensional feature.

results. For Adware and Dropper malware, the highest detection accuracy is achieved with 5-dimensional and 7-dimensional features, respectively. Lastly, 7-dimensional features are most effective for detecting Virus malware.

In our next evaluation, we assessed the impact of using unbalanced samples versus applying the SMOTE sample balancing algorithm to balance sample distributions. As shown in Table 7, the application of the SMOTE algorithm consistently improves the overall performance of our model, irrespective of whether we use 1-dimensional, 3-dimensional, 5-dimensional, or 7-dimensional features. This improvement is further demonstrated in Fig 10, where we evaluated Dataset 2 using a model trained on 5-dimensional node features. By comparing the loss curves of models trained with and without SMOTE, we observe that the samples processed using the SMOTE algorithm exhibit lower training error. These findings

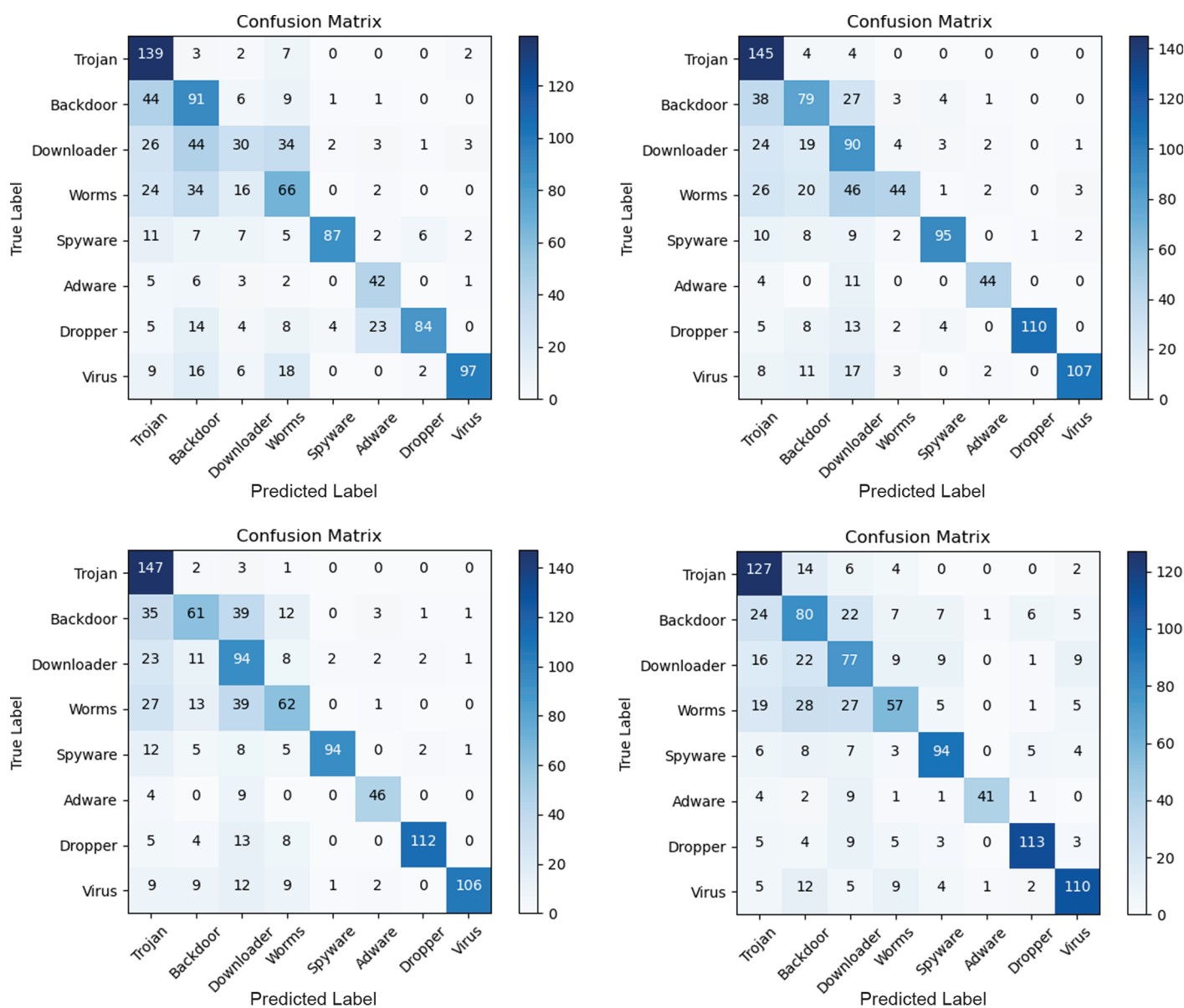

**Fig 9. Confusion matrix performance of multi-dimensinal feature in dataset2.** (a): Confusion matrix performance of 1-dimensional feature. (b): Confusion matrix performance of 3-dimensional feature. (c):Confusion matrix performance of 5-dimensional feature. (d):Confusion matrix performance of 7-dimensional feature.

collectively indicate that our model significantly benefits from the SMOTE sample balancing algorithm, which improves its performance in malware classification.

**Analysis of the algorithm's time complexity.** In the FSGCN operation, this paper only uses a single layer of GCN. According to Eq (1), the time complexity of a single layer of GCN is $O\left(|E|D + N3D^2\right)$, where $|E|$ represents the number of edges, $N$ is the number of nodes in the graph, and $D$ is the feature dimension of the nodes.

Next, in the CNN classification operation, we use two convolutional layers and three fully connected layers. Suppose the input feature map of the $l$-th layer has a size of $H_l \times W_l$ (height

**Table 7. Ablation analysis of our method on Dataset 2 With and without SMOTE.**

| | w/o SMOTE | | | | w/ SMOTE | | | |
|---|---|---|---|---|---|---|---|---|
| **Feature number** | *Accuracy* | *Recall* | *Precision* | *F1* | *Accuracy* | *Recall* | *Precision* | *F1* |
| 1-dimensional feature | 0.576 | 0.576 | 0.631 | 0.568 | 0.597 | 0.597 | 0.640 | 0.595 |
| 3-dimensional feature | 0.652 | 0.652 | 0.688 | 0.646 | 0.670 | 0.670 | 0.730 | 0.672 |
| 5-dimensional feature | 0.645 | 0.645 | 0.680 | 0.652 | 0.676 | 0.676 | 0.713 | 0.675 |
| 7-dimensional feature | 0.616 | 0.616 | 0.657 | 0.629 | 0.656 | 0.656 | 0.670 | 0.656 |

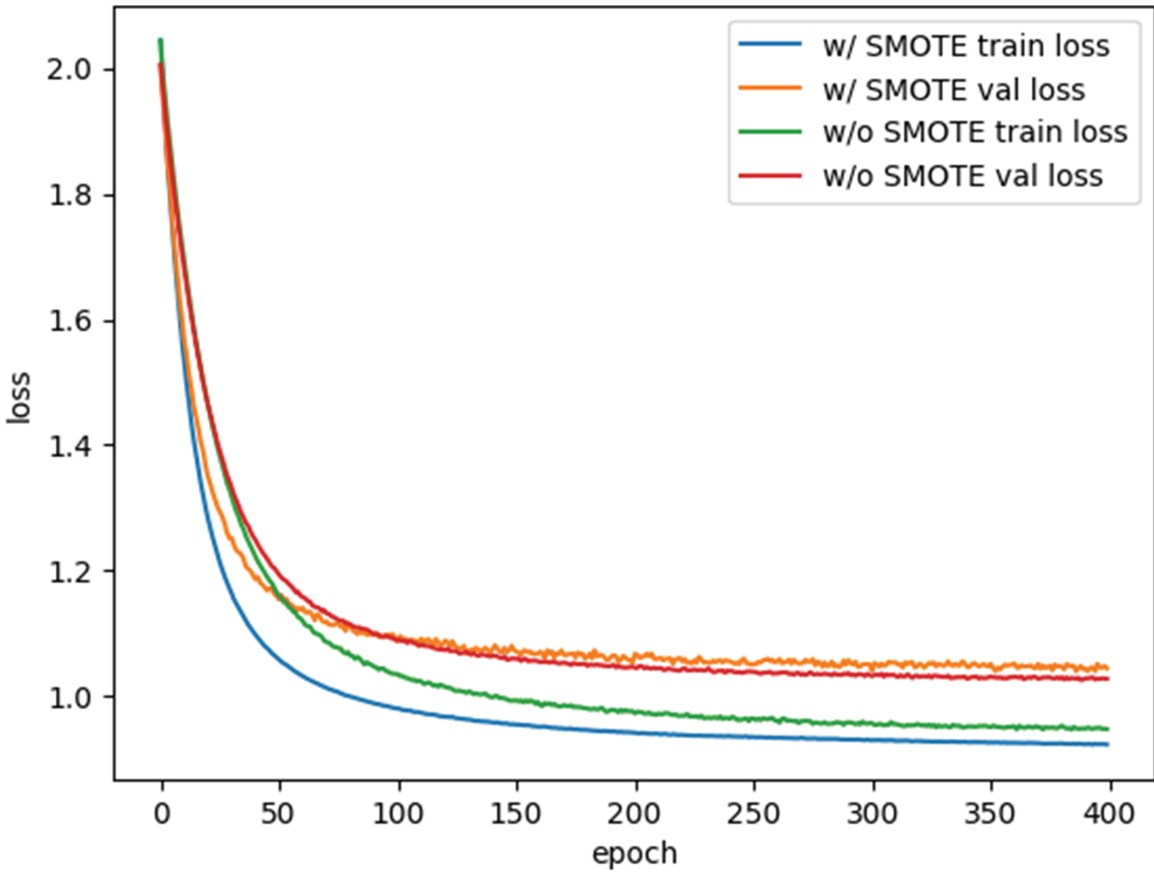

**Fig 10. Comparison of loss curves for dataset 2 with 5-dimensional features: w/o SMOTE vs. w/ SMOTE.**

and width), and the convolution kernel size is $k_l \times k_l$. The time complexity of the two convolutional layers is: $O\left(\sum_{l=1}^{2} H_{l+1} \times W_{l+1} \times k_l^2\right)$. The time complexity of the three fully connected layers is: $O\left(\sum_{i=1}^{3} N_i \times N_{i+1}\right)$, where $N_i$ is the number of input neurons of the $i$-th fully connected layer, and $N_{i+1}$ is the number of output neurons of the $i$-th fully connected layer. The final overall time complexity is: $O\left(|E| D + N3D^2\right) + O\left(\sum_{l=1}^{2} H_{l+1} \times W_{l+1} \times k_l^2\right) + O\left(\sum_{i=1}^{3} N_i \times N_{i+1}\right)$.

## Conclusion

This research makes significant advancements in malware classification by addressing the limitations of existing methods that utilize API sequences. Many current approaches overlook the structural information within API sequences, and although some methods consider the sequence structure, they treat the sequences as unordered, disregarding the inherent sequential relationships between API calls. This oversight leads to suboptimal classification performance.

To address these challenges, we propose a malware classification method based on directed API call relationships. This approach models the API sequence as a directed graph, incorporating node attributes, structural, and directional information. To extract these features, we leverage the principles of GCN by constructing first- and second-order convolutional neural networks to approximate the convolution operations on the directed graph. The resulting API sequence representation is converted into a grayscale image and classified using a CNN. We also apply the SMOTE algorithm to address class imbalance in the datasets.

These approaches result in significant performance improvements on two datasets. In future work, we plan to explore additional datasets, experiment with different architectures, and further enhance our model to improve its generalizability and robustness.

## Author contributions

**Conceptualization:** Cuihua Ma, Anas Bilal, Xiaowen Liu.

**Formal analysis:** Haixia Long.

**Funding acquisition:** Haixia Long.

**Methodology:** Cuihua Ma, Zhenwan Li, Anas Bilal.

**Project administration:** Haixia Long.

**Resources:** Xiaowen Liu.

**Software:** Cuihua Ma, Anas Bilal.

**Supervision:** Haixia Long.

**Validation:** Zhenwan Li, Xiaowen Liu.

**Visualization:** Zhenwan Li.

**Writing – original draft:** Cuihua Ma.

**Writing – review & editing:** Anas Bilal.

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
