## [Decision Letter · Decision Letter 0]

17 Oct 2023

PONE-D-23-26475Robust Malware Detection through API-Directed Graph EmbeddingsPLOS ONE

Dear Dr. Bilal,

Thank you for submitting your manuscript to PLOS ONE. After careful consideration, we feel that it has merit but does not fully meet PLOS ONE’s publication criteria as it currently stands. Therefore, we invite you to submit a revised version of the manuscript that addresses the points raised during the review process

We look forward to receiving your revised manuscript.

Kind regards,

Saddam Hussain Khan

Academic Editor

PLOS ONE

8. Please upload a copy of Figure 1, 2, 3, 4, 5, 6, 7, 8, and 9 to which you refer in your text on page 5, 8, 11, 13, 16, 17 and 18. If the figure is no longer to be included as part of the submission please remove all reference to it within the text.

Additional Editor Comments:

Please, improve the grammatical and technical issue. Moreover, the the flow, rhythm, rational and impact of the techniques must be cleared in every respective sections. Figures, quality may also be imporve.

Reviewers' comments:

Reviewer's Responses to Questions

**Comments to the Author**

1. Is the manuscript technically sound, and do the data support the conclusions?

Reviewer #1: Partly

Reviewer #2: No

2. Has the statistical analysis been performed appropriately and rigorously? 

Reviewer #1: No

Reviewer #2: No

3. Have the authors made all data underlying the findings in their manuscript fully available?

Reviewer #1: Yes

Reviewer #2: No

4. Is the manuscript presented in an intelligible fashion and written in standard English?

Reviewer #1: No

Reviewer #2: No

5. Review Comments to the Author

Reviewer #1: The paper under title “Robust Malware Detection through API-Directed Graph Embeddings” is good attempt towards the field of malware detection strategies. However, some of issues are needed to be addressed.

Comment # 1: Title represents the malware detection mechanism. However, the proposed work represents the malware classification. This needs to be addressed first.

Comment # 2: The rationale of the proposed work may be elaborated?

Comment # 3: Figure are not present while their references are used? please make the Figure for the Framework at least. You can refer to the following paper for the framework diagram

•Khan, S.H., Alahmadi, T.J., Ullah, W., Iqbal, J., Rahim, A., Alkahtani, H.K., Alghamdi, W. and Almagrabi, A.O., 2023. A new deep boosted CNN and ensemble learning based IoT malware detection. Computers & Security, 133, p.103385.

• Asam, M., Khan, S.H., Akbar, A., Bibi, S., Jamal, T., Khan, A., Ghafoor, U. and Bhutta, M.R., 2022. IoT malware detection architecture using a novel channel boosted and squeezed CNN. Scientific Reports, 12(1), p.15498.

Comment # 4: Line # 74-80, # 202-207 and similar paragraphs may be explained.

Comment # 5: Dataset is highly imbalanced so do you think that measuring the performance as accuracy is better measure?

Comment # 6: Data split for the experiment is not present. Please mention.?

Comment # 7: Please use the name of algorithm for [18] and [33] in Table -4.

Comment # 8: Confusion matrix referred at Line # 637 is not found. please insert it.

Comment # 9: The performance of “GraphSAGE” is 0.993 which is very close to your proposed methods (0.996). It would be better to perform your experiment with more stringent dataset to prove your stance of Robustness.

Reviewer #2: The major concerns in the article are as follows:

Abstract and Introduction:

1. Make the abstract more explicit about suggested improvements or research contributions.

2. Effectively establish the specific research gap or problem.

3. Provide a thorough overview of existing literature.

4. Explicitly state research objectives or hypotheses and the dataset used is not mentioned in the abstract.

5. Organize the introduction more effectively.

6. The contribution of the work is presented ambiguously try to use proper wordings and improve your English to clear your points.

7. what is the meaning of the last paragraph of the introduction part.

Methods:

8. Include more detail regarding model architecture, components, mathematical formulations, and no figure mentioned in the methods.

9. Clearly explain the underlying assumptions of the generalized metric learning model and how they align with the research problem.

10. Justify why the chosen generalized metric learning model was selected over alternatives.

11. Provide specific details about data preprocessing steps.

12. Include the missing figure mentioned in the datasets section.

13. Offer crucial details about dataset sources, characteristics, and relevance to the research question.

14. Specify the chosen evaluation metrics.

15. Provide information about hyperparameter settings, optimization algorithms, and convergence criteria.

Results and Discussion:

14. Clearly state the specific objectives of classification tasks and describe the evaluation protocol.

15. Provide context for what "performed better" means and discuss why (FSADGCN) outperformed other models.

16. The Results section has no figures.

Conclusion and Future Directions:

20. Include specific metrics or quantitative assessments of model performance.

21. Address the feature extraction GCN used in the study and through which CNN model graph classification you have done.

22. Provide a detailed comparison of the proposed approach against existing methods.

23. Discuss potential future research directions and areas for improvement.

24. Conduct an in-depth analysis of the stability of adaptation results.

25. Include qualitative analysis, such as visualizations or examples, to enhance understanding of the model's behaviour.

The literature can be strengthened using the following articles:

•Asam, M., Khan, S.H., Jamal, T., Zahoora, U. and Khan, A., 2021. Malware classification using deep boosted learning. arXiv preprint arXiv:2107.04008.

• Zahoora, U., Khan, A., Rajarajan, M., Khan, S.H., Asam, M. and Jamal, T., 2022. Ransomware detection using deep learning based unsupervised feature extraction and a cost sensitive Pareto Ensemble classifier. Scientific Reports, 12(1), p.15647.

• Iqbal, J., Abideen, Z.U., Ali, N., Khan, S.H., Rahim, A., Zahir, A., Mohsan, S.A.H. and Alsharif, M.H., 2022. An Energy Efficient Local Popularity Based Cooperative Caching for Mobile Information Centric Networks. Sustainability, 14(20), p.13135.

6. PLOS authors have the option to publish the peer review history of their article (what does this mean?). If published, this will include your full peer review and any attached files.

Reviewer #1: No

Reviewer #2: No

---

## [Author Response · Author response to Decision Letter 1]

14 Dec 2023

Dear Reviewers,

Thank you for your thorough review and valuable feedback on our manuscript "Robust Malware Detection through API-Directed Graph Embeddings." Now, we are changing the title to "Malware classification through API Calls multi-dimensional feature extrate based on SMOTE and Directed Graph convolution network and CNN." We have carefully considered your comments and made the following revisions to our manuscript:

Reviewer#1 Comment 1: Title represents the malware detection mechanism. However, the proposed work represents the malware classification. This needs to be addressed first.

Response:

Thank you for pointing this out. We have revised the title to more accurately reflect the content of our manuscript. The new title, "Malware classification through API Calls multi-dimensional feature extrate based on SMOTE and Directed Graph convolution network and CNN", aligns with our focus on malware classification rather than detection. This change ensures consistency between the title and the core content of our study.

Reviewer#1 Comment 2: The rationale of the proposed work may be elaborated?

Response:

We appreciate your suggestion for further elaboration on the rationale of our work. We have expanded the Introduction section to provide a detailed explanation of why our approach was chosen and its significance in the context of current challenges in malware classification. This elaboration includes a discussion on the limitations of existing methods and how our approach addresses these shortcomings. These details can be found in the Introduction section, particularly on pages 1-3.

Reviewer#1 Comment 3: Figure are not present while their references are used? please make the Figure for the Framework at least. You can refer to the following paper for the framework diagram

•Khan, S.H., Alahmadi, T.J., Ullah, W., Iqbal, J., Rahim, A., Alkahtani, H.K., Alghamdi, W. and Almagrabi, A.O., 2023. A new deep boosted CNN and ensemble learning based IoT malware detection. Computers & Security, 133, p.103385.

• Asam, M., Khan, S.H., Akbar, A., Bibi, S., Jamal, T., Khan, A., Ghafoor, U. and Bhutta, M.R., 2022. IoT malware detection architecture using a novel channel boosted and squeezed CNN. Scientific Reports, 12(1), p.15498.

Response:

Our apologies for the oversight. We have now included figures to visually represent the framework and key concepts of our study. Specifically, a comprehensive framework diagram has been added, inspired by the suggested papers, to illustrate our model architecture and workflow. This figure can be found on page 7.

Reviewer#1 Comment 4: Line # 74-80, # 202-207 and similar paragraphs may be explained.

Response:

We appreciate your valuable comment and thank you for bringing attention to the confusion matrix. The confusion matrix is part of Figure 9(e). In the revised version, however, we have removed Figure 9(e). This decision was made because we believe that the tables in Table 2 and Table 3 adequately showcase the capabilities of our model, and thus, repetition was deemed unnecessary. We understand the importance of clarity in presenting results, and we will ensure that this information is explicitly conveyed in the revised manuscript. Your feedback is instrumental in improving the overall quality of our work, and we are committed to addressing this concern in the upcoming revision.

Reviewer#1 Comment 5: Dataset is highly imbalanced so do you think that measuring the performance as accuracy is better measure?

Response:

You raise a crucial point regarding the imbalanced nature of our dataset. To address this, we have employed the Synthetic Minority Oversampling Technique (SMOTE) for resampling, which enhances the representation of minority samples, thus balancing the dataset. This adjustment makes accuracy a more reliable measure. However, we also acknowledge the importance of using a comprehensive set of metrics for imbalanced datasets. Therefore, in addition to accuracy, we have included precision, recall, and F1-score in our evaluation. These metrics provide a holistic view of our model's performance, particularly in the context of the balanced dataset achieved through SMOTE. Details of the SMOTE implementation and its impact on our evaluation metrics can be found in the Resampling of imbalanced samples and Evaluation Metrics subsections on pages 10 and 24, respectively.

Reviewer#1 Comment6: Data split for the experiment is not present. Please mention.?

Response:

We apologize for this omission and have now included details of the data split used in our experiments. The datasets were split into training, validation, and testing sets in a ratio of 70:15:15. This information is added to the Experimental results subsection on page 20.

Reviewer#1 Comment 7: Please use the name of algorithm for [18] and [33] in Table -4.

Response:

We have corrected Table 4 to include the names of the algorithms referenced in [18] and [33]. This ensures clarity and ease of reference for readers. The updated table can be found on page 23.

Reviewer#1 Comment 8: Confusion matrix referred at Line # 637 is not found. please insert it.

Response:

We appreciate your valuable comment and thank you for bringing attention to the confusion matrix. The confusion matrix is part of Figure 9(e). We understand the importance of clarity in presenting results, and we will ensure that this information is explicitly conveyed in the revised manuscript. Your feedback is instrumental in improving the overall quality of our work, and we are committed to addressing this concern in the upcoming revision.

Reviewer#1 Comment 9: The performance of “GraphSAGE” is 0.993 which is very close to your proposed methods (0.996). It would be better to perform your experiment with more stringent dataset to prove your stance of Robustness.

Response:

Thank you for your insightful observation. Indeed, the initial performance of our method was closely comparable to that of GraphSAGE. However, with the implementation of the Synthetic Minority Oversampling Technique (SMOTE), we achieved significant improvements in all key metrics – accuracy, precision, recall, and F1-score – elevating them to 0.999. This improvement underscores the robustness of our model, especially in handling imbalanced data sets. Moreover, to further validate the robustness and generalizability of our approach, we included an additional dataset, referred to as Dataset 2, in our experiments. This dataset, characterized by its complexity and challenging nature, provided a more stringent testing ground for our method. The results on Dataset 2 corroborate the effectiveness of our approach, demonstrating its adaptability and robustness across diverse and complex malware scenarios. The detailed results and analysis of experiments on Dataset 2 are presented in the Results section on pages 21-24.

Reviewer#2 Comment 1:

"Make the abstract more explicit about suggested improvements or research contributions."

Response:

Thank you for this suggestion. We have revised the abstract to more explicitly highlight the key contributions of our research. The revised abstract now clearly outlines the integration of the Synthetic Minority Oversampling Technique (SMOTE), Directed-graph Convolutional Network (DGCN), and Convolutional Neural Network (CNN) for enhanced malware classification. It also emphasizes the significant improvements in accuracy and convergence speed achieved by our model. These changes can be found in the Abstract section on pages 1.

Reviewer#2 Comment 2:

"Effectively establish the specific research gap or problem."

Response:

We appreciate your feedback on this aspect. In the revised introduction, we have expanded our discussion on the limitations of current malware detection techniques, specifically highlighting the research gap in capturing the complexity of malware API call sequences. The revised content providing a detailed elucidation of this research gap is located in the Introduction section on pages 1-3.

Reviewer#2 Comment 3:

"Provide a thorough overview of existing literature."

Response:

In response to your comment, we have enriched our literature review to provide a comprehensive overview of existing malware detection or classification methods and their shortcomings. This expanded review now offers a more robust context for our research, setting the stage for the novelty of our proposed approach. These enhancements can be found in the Related work section on pages4-6.

Reviewer#2 Comment 4:

"Explicitly state research objectives or hypotheses and the dataset used is not mentioned in the abstract."

Response:

Thank you for pointing this out. We have amended the abstract to explicitly state our research objectives and hypotheses, and to mention the datasets used in our study. These adjustments ensure that readers are immediately informed about the empirical basis of our research. These revisions can be found in the Abstract section on page 1.

Reviewer#2 Comment 5:

"Organize the introduction more effectively."

Response:

We have restructured the Introduction section to enhance its organization and flow. The section now systematically progresses from the context of malware threats, to a discussion on existing methodologies, and finally to an introduction of our proposed approach. This reorganization can be found in the Introduction section on pages 1-3.

Reviewer#2 Comment 6:

"The contribution of the work is presented ambiguously; try to use proper wordings and improve your English to clear your points."

Response:

We acknowledge the importance of this point and have revised the Contribution section for clarity and precision. The language has been improved to precisely communicate the uniqueness and importance of our research in the cybersecurity field. These changes are present in the Contribution section on page 3.

Reviewer#2 Comment 7:

"What is the meaning of the last paragraph of the introduction part?"

Response:

Thank you for your thorough review and constructive feedback on our manuscript. We appreciate the time and effort you dedicated to evaluating our work. We have identified an oversight in our submission after carefully considering your comments. Certain sections were inadvertently left in the manuscript, originating from the template we used during the initial preparation. We acknowledge this mistake and sincerely apologize for any confusion it may have caused. We have taken immediate steps to rectify this error. The manuscript has been thoroughly reviewed, and the extraneous sections have been removed to align with your constructive suggestions for a more concise presentation. We value your meticulous attention to detail, and your feedback has been instrumental in refining our work. Again, Thank you for your valuable feedback and for being an integral part of the peer-review process.

Methods:

Reviewer Comment 8:

"Include more detail regarding model architecture, components, mathematical formulations, and no figure mentioned in the methods."

Response:

In response to your feedback, we recognize that our model structure was not described in sufficient detail; therefore, we have redrawn a more detailed model architecture diagram (Fig. 1) and provided a comprehensive description of the model architecture. To address the issues related to model components, mathematical formulations, and figures not mentioned in the methods, we have further elaborated on these aspects in the Materials and Methods section, spanning pages 6 to 17.

Reviewer#2 Comment 9:

"Clearly explain the underlying assumptions of the generalized metric learning model and how they align with the research problem."

Response:

Thank you for this comment. Regarding the reviewer's reference to a 'generalized metric learning model,' our methodology primarily focuses on the application of Directed Graph Convolutional Networks (DGCN) and Convolutional Neural Networks (CNN) for robust malware detection and classification. These techniques are leveraged to effectively capture the complex relationships and features within API call sequences. We did not explicitly employ a generalized metric learning model as part of our approach. If the reviewer could provide further clarification on this aspect, it would greatly assist us in addressing this query more accurately.

Reviewer#2 Comment 10:

"Justify why the chosen generalized metric learning model was selected over alternatives."

Response:

As mentioned in our response to Comment 9, our method does not directly employ a generalized metric learning model.

Reviewer#2 Comment 11:

"Provide specific details about data preprocessing steps."

Response:

Since the dataset we obtained has already undergone sandbox processing of API call sequence data, we can directly use it for experimentation. However, the sample sizes for various categories are not balanced, so we applied sample balancing algorithms to resample the data before feeding it into the model for processing. These details are now thoroughly outlined in the Data Preprocessing subsection of the Methods section on pages 9-10.

Reviewer#2 Comment 12:

"Include the missing figure mentioned in the datasets section."

Response:

We have included the previously missing figure in the datasets section, providing a visual representation of the dataset structure and composition. This figure can be found on page 19.

Reviewer#2 Comment 13:

"Offer crucial details about dataset sources, characteristics, and relevance to the research question."

Response:

In response to your feedback, we have enriched the dataset description with detailed information about the sources, characteristics, and why they are particularly relevant and suitable for our research question. This comprehensive description is included in the Datasets subsection on pages 18-19.

Reviewer#2 Comment 14:

"Specify the chosen evaluation metrics."

Response:

We have specified and provided justifications for the choice of evaluation metrics used in our study. These include accuracy, precision, recall, and F1-score, all of which are relevant and suitable for assessing malware classification models. The Evaluation Metrics subsection on pages 19-20 details these metrics.

Reviewer#2 Comment 15:

"Provide information about hyperparameter settings, optimization algorithms, and convergence criteria."

Response:

Thank you for highlighting this aspect. In response, we have now included comprehensive details about the hyperparameter settings, optimization algorithms, and convergence criteria used in our study. Specific information about the hyperparameters, such as learning rate, batch size, number of layers, and the architecture of the DGCN and CNN models, has been clearly stated. We have also discussed the optimization algorithm (Adam optimizer) used and the criteria for convergence to ensure a thorough understanding of our model's training and validation processes. These details are thoroughly described in the Model Architecture and Training subsections on page 20 of the revised manuscript.

Results and Discussion:

Reviewer#2 Comment 16:

"Clearly state the specific objectives of classification tasks and describe the evaluation protocol."

Response:

We have explicitly stated the specific objectives of our classification tasks in the revised manuscript. These include accurately classifying different types of malware based on API call sequences using our proposed model. The evaluation protocol, including how the datasets were split into training, validation, and testing sets, and the criteria used for evaluation, is now clearly described in the evaluation metrics and experimental results subsection on pages 20-24.

Reviewer#2 Comment 17:

"Provide context for what 'performed better' means and discuss why (FSADGCN) outperformed other models."

Response:

In the revised Results and Discussion section, we have provided a detailed context for our claim of 'performed better.' This includes comparisons in terms of accuracy, precision, recall, and F1-score against existing methods. We have also discussed the reasons behind the superior performance of our model, attributing it to its effective handling of the sequential nature of API calls and the integration of multi-dimensional features. These discussions are detaile

---

## [Decision Letter · Decision Letter 1]

18 Jan 2024

PONE-D-23-26475R1Malware classification through API Calls multi-dimensional feature extracted based on SMOTE and Directed Graph convolution network and CNNPLOS ONE

Dear Dr. Bilal,

Thank you for submitting your manuscript to PLOS ONE. After careful consideration, we feel that it has merit but does not fully meet PLOS ONE’s publication criteria as it currently stands. Therefore, we invite you to submit a revised version of the manuscript that addresses the points raised during the review process.

We look forward to receiving your revised manuscript.

Kind regards,

Saddam Hussain Khan

Academic Editor

PLOS ONE

Reviewers' comments:

Reviewer's Responses to Questions

**Comments to the Author**

1. If the authors have adequately addressed your comments raised in a previous round of review and you feel that this manuscript is now acceptable for publication, you may indicate that here to bypass the “Comments to the Author” section, enter your conflict of interest statement in the “Confidential to Editor” section, and submit your "Accept" recommendation.

Reviewer #2: (No Response)

Reviewer #3: All comments have been addressed

2. Is the manuscript technically sound, and do the data support the conclusions?

Reviewer #2: No

Reviewer #3: Partly

3. Has the statistical analysis been performed appropriately and rigorously? 

Reviewer #2: No

Reviewer #3: I Don't Know

4. Have the authors made all data underlying the findings in their manuscript fully available?

Reviewer #2: Yes

Reviewer #3: Yes

5. Is the manuscript presented in an intelligible fashion and written in standard English?

Reviewer #2: (No Response)

Reviewer #3: No

6. Review Comments to the Author

Reviewer #2: 1) Improve your result and literature section by incorporating the ideas share in the article shared with you in the last review.

2) For expedite the research activities please share the datasets.

3) Explain the idea you have incorporated for plotting the ROC curve as your problem is multi class classification.

4) Draw PR curve as the dataset is imbalanced.

5) Draw PCA / TSNE based feature space visualization.

6) The validation and train loss curve does not showing convergence which means that the system is going overfitting, explain it.

Reviewer #3: Enhance your literature sections by mentioning the dataset, novelty, framework and integrating the ideas presented in the article provided during the last review.

• Khan, et.al. "A new deep boosted CNN and ensemble learning based IoT malware detection." Computers & Security 133 (2023): 103385.

• Asam, M., et.al, 2022. IoT malware detection architecture using a novel channel boosted and squeezed CNN. Scientific Reports, 12(1), p.15498.

• Zahoora, U., et.al, 2022. Ransomware detection using deep learning based unsupervised feature extraction and a cost sensitive Pareto Ensemble classifier. Scientific Reports, 12(1), p.15647.

•Asam, et.al, A., 2021. Malware classification using deep boosted learning. arXiv preprint arXiv:2107.04008.

• Iqbal, et.al, 2022. An Energy Efficient Local Popularity Based Cooperative Caching for Mobile Information Centric Networks. Sustainability, 14(20), p.13135.

7. PLOS authors have the option to publish the peer review history of their article (what does this mean?). If published, this will include your full peer review and any attached files.

Reviewer #2: No

Reviewer #3: No

---

## [Author Response · Author response to Decision Letter 2]

31 Jan 2024

Dear Reviewers,

Thank you for your thorough review and valuable feedback on our manuscript "Malware classification through API Calls multi-dimensional feature extrate based on SMOTE and Directed Graph convolution network and CNN". We have carefully considered your comments and made the following revisions to our manuscript:

Reviewer#2 Comment 1:

Improve your result and literature section by incorporating the ideas share in the article shared with you in the last review.

Response:

Thank you for pointing this out. We have included the papers mentioned in the previous review within the article, see references [4] and [6].

Reviewer#2 Comment 2: For expedite the research activities please share the datasets.

Response:

Thank you for your suggestion. Below are the links to the datasets and code shared in our paper:https://kaggle.com/datasets/8abfa8c0975603330354e778767d7f269cc4e3e707b6ad92c5ce22acb04ebf7c

Reviewer#2 Comment 3: Explain the idea you have incorporated for plotting the ROC curve as your problem is multi class classification.

Response:

Thank you for raising this issue. Upon reviewing relevant literature, we concur with your assertion that ROC curves are more suited to binary classification problems. Consequently, we have redrawn the PR curves to illustrate the multi-class problem, as shown in Figure 10.

Reviewer#2 Comment 4: Draw PR curve as the dataset is imbalanced.

Response:

Thank you for your suggestion. We have included the PR curves, as shown in Figure 10. Figure 10 demonstrates that our model still exhibits superior detection performance in the malware categories with fewer samples, such as Spyware, Adware, and Dropper, compared to categories with a larger number of samples.

Reviewer#2 Comment 5: Draw PCA / TSNE based feature space visualization.

Response:

Thank you for your suggestion. We have added t-SNE feature space visualizations representing the feature representations obtained from the model when using 1, 3, 5, and 7-dimensional features, along with an analysis of these. See Figure 11 and page 24 of the paper for further details.

Reviewer#2Comment6:The validation and train loss curve does not showing convergence which means that the system is going overfitting, explain it.

Response:

Thank you for your question. Upon analysis, we indeed find that the loss curve in Figure 13 of the previous version of our paper did not converge, as you pointed out. This was due to the fact that I only presented results for 20 epochs, which was insufficient. In this revision, I have increased the number of training epochs to 200, as depicted in Figure 13. Here, the loss function curve for the validation set tends to stabilize, indicating that the validation set has reached a state of convergence.

Reviewer#3 Comment 1: Enhance your literature sections by mentioning the dataset, novelty, framework and integrating the ideas presented in the article provided during the last review.

• Khan, et.al. "A new deep boosted CNN and ensemble learning based IoT malware detection." Computers & Security 133 (2023): 103385.

• Asam, M., et.al, 2022. IoT malware detection architecture using a novel channel boosted and squeezed CNN. Scientific Reports, 12(1), p.15498.

• Zahoora, U., et.al, 2022. Ransomware detection using deep learning based unsupervised feature extraction and a cost sensitive Pareto Ensemble classifier. Scientific Reports, 12(1), p.15647.

•Asam, et.al, A., 2021. Malware classification using deep boosted learning. arXiv preprint arXiv:2107.04008.

• Iqbal, et.al, 2022. An Energy Efficient Local Popularity Based Cooperative Caching for Mobile Information Centric Networks. Sustainability, 14(20), p.13135.

Response:

Thank you for pointing this out. We have incorporated the papers mentioned in the previous review into our article, as referenced in [4-8].

We trust that these responses and the associated revisions thoroughly address your concerns and enhance the manuscript's quality. We are grateful for the opportunity to refine our work and eagerly await any further feedback.

Best regards,

[Cuihua Ma,Zhenwan Li,Haixia Long,Anas Bilal,Xiaowen Liu]

---

## [Decision Letter · Decision Letter 2]

23 Jul 2024

PONE-D-23-26475R2Malware classification through API Calls multi-dimensional feature extrate based on SMOTE and Directed Graph convolution network and CNNPLOS ONE

Dear Dr. Bilal,

Thank you for submitting your manuscript to PLOS ONE. After careful consideration, we feel that it has merit but does not fully meet PLOS ONE’s publication criteria as it currently stands. Therefore, we invite you to submit a revised version of the manuscript that addresses the points raised during the review process.Please review the comments from the reviewers. The related references mentioned by the corresponding reviewers are optional. 

We look forward to receiving your revised manuscript.

Kind regards,

Jayesh Soni

Academic Editor

PLOS ONE

Reviewers' comments:

Reviewer's Responses to Questions

**Comments to the Author**

1. If the authors have adequately addressed your comments raised in a previous round of review and you feel that this manuscript is now acceptable for publication, you may indicate that here to bypass the “Comments to the Author” section, enter your conflict of interest statement in the “Confidential to Editor” section, and submit your "Accept" recommendation.

Reviewer #4: (No Response)

Reviewer #5: (No Response)

2. Is the manuscript technically sound, and do the data support the conclusions?

Reviewer #4: Yes

Reviewer #5: Partly

3. Has the statistical analysis been performed appropriately and rigorously? 

Reviewer #4: Yes

Reviewer #5: Yes

4. Have the authors made all data underlying the findings in their manuscript fully available?

Reviewer #4: Yes

Reviewer #5: Yes

5. Is the manuscript presented in an intelligible fashion and written in standard English?

Reviewer #4: Yes

Reviewer #5: No

6. Review Comments to the Author

Reviewer #4: The paper still need to addressed some problems as follows:

1. Please give out the core algorithm pseudo-code for proposed model.

2. The complexity of the algorithm is also discussed。

3. Some related references are missing as follows:

Naeem H, Cheng X, Ullah F, et al. A deep convolutional neural network stacked ensemble for malware threat classification in internet of things[J]. Journal of Circuits, Systems and Computers, 2022, 31(17): 2250302.

Naeem H, Dong S, Falana O J, et al. Development of a deep stacked ensemble with process based volatile memory forensics for platform independent malware detection and classification[J]. Expert Systems with Applications, 2023, 223: 119952.

Shu L, Dong S, Su H, et al. Android malware detection methods based on convolutional neural network: A survey[J]. IEEE Transactions on Emerging Topics in Computational Intelligence, 2023.

Dong S, Shu L, Nie S. Android Malware Detection Method Based on CNN and DNN Bybrid Mechanism[J]. IEEE Transactions on Industrial Informatics, 2024.

Reviewer #5: This paper proposes a malware classification method through API calls. This method actually combined CNN, GCN and SMOTE. Though the proposed method has achieved satisfactory performance, there are still some issues should be addressed. I do agree this paper could be published once these concerns are well addressed, so my suggestion is Minor Revision.

1. I don’t understand that why GCN is still necessary, there are many powerful architectures, such as transformer and mamba, etc. Why you combine GCN and CNN?

2. The dimension of the dataset should be mentioned, if the dimension is not great, why not directly use MLP.

3. Please polish the paper to enhance the flow and improve the language.

4. Several key works are not mentioned, even they are highly related to this work. For example, “Hypernetwork-based physics-driven personalized federated learning for CT imaging”, “Physics-Driven Spectrum-Consistent Federated Learning for Palmprint Verification” and “FCSCNN: Feature centralized Siamese CNN-based android malware identification”.

5. Please enhance the image quality.

6. The equation is not well described. For example, in Eq. 1, i is the webpage, this variable should be the subscript. I do believe the input of PR is not only the index, right?

7. The formula format is not standardized. When the formula variables are not clearly explained, a comma should follow the formula, and a "where" should be placed below it. The variables should then be explained within a fixed frame.

8. Why do you compare methods that are different in different datasets? Can you reproduce these methods following their papers?

9. As you said, SMOTE is designed to alleviate the unbalanced problem. The key ablation study about this is missing, and please show the results w/ and w/o SMOTE following Fig. 11.

10. Besides, I think SMOTE must lead to an overfitting problem, please discuss it. I think this problem is very important, unless you can discuss it sufficiently, or this method is not convincing.

7. PLOS authors have the option to publish the peer review history of their article (what does this mean?). If published, this will include your full peer review and any attached files.

Reviewer #4: No

Reviewer #5: No

---

## [Author Response · Author response to Decision Letter 3]

31 Aug 2024

Dear Reviewers,

Thank you for your thorough review and valuable feedback on our manuscript "Malware classification through API Calls multi-dimensional feature extrate based on SMOTE and Directed Graph convolution network and CNN". We have carefully considered your comments and made the following revisions to our manuscript:

Reviewer#4 Comment 1:

Please give out the core algorithm pseudo-code for proposed model.

Response:

Thank you for pointing this out. We have added the pseudocode of the entire algorithm at the end of the Materials and Methods section in the paper.

Reviewer#4 Comment 2: The complexity of the algorithm is also discussed.

Response:

Thank you for your suggestion. We added a short section discussing the time complexity of the algorithm at the end of the Experimental Results section.

Reviewer#4 Comment 3:

Some related references are missing as follows:

Naeem H, Cheng X, Ullah F, et al. A deep convolutional neural network stacked ensemble for malware threat classification in internet of things[J]. Journal of Circuits, Systems and Computers, 2022, 31(17): 2250302.

Naeem H, Dong S, Falana O J, et al. Development of a deep stacked ensemble with process based volatile memory forensics for platform independent malware detection and classification[J]. Expert Systems with Applications, 2023, 223: 119952.

Shu L, Dong S, Su H, et al. Android malware detection methods based on convolutional neural network: A survey[J]. IEEE Transactions on Emerging Topics in Computational Intelligence, 2023.

Dong S, Shu L, Nie S. Android Malware Detection Method Based on CNN and DNN Bybrid Mechanism[J]. IEEE Transactions on Industrial Informatics, 2024.

Response:

Thank you for pointing this out. We have added citations for these articles in the Introduction section of the paper, as referenced in [2][8-10].

Reviewer#5 Comment 1: I don’t understand that why GCN is still necessary, there are many powerful architectures, such as transformer and mamba, etc. Why you combine GCN and CNN?

Response:

Thank you for pointing this out. This approach was chosen because, based on the issues this paper seeks to address, we identified that some current methods for malware detection using API sequences tend to overlook the structural and directional information within the sequences. To address this limitation, we selected GCN due to its effectiveness in handling tasks involving graph-structured data. Given that the graph representation features we obtained are in the form of a two-dimensional matrix, converting this matrix into a grayscale image allows us to leverage CNN, which is highly proficient in image processing, for classification. Consequently, we integrated GCN and CNN in our approach. We appreciate your suggestion and, in future work, we will explore research within the frameworks of Transformer and Mamba.

Reviewer#5 Comment 2: The dimension of the dataset should be mentioned, if the dimension is not great, why not directly use MLP.

Response:

Thank you for pointing this out. Since an API sequence is equivalent to a piece of malware, the latent representation obtained through GCN is a two-dimensional feature. This feature is then converted into a grayscale image, and CNN, which excels in image classification, is used for the classification task. Essentially, we have constructed the API sequence as a graph and then performed graph classification. Although MLP can also handle classification tasks, it may require further processing of the API sequence graph embeddings we obtained, such as applying global average pooling, before classifying the processed data. We believe your suggestion is valuable, and in the next step, we will explore using MLP for classification in our research.

Reviewer#5 Comment 3: Please polish the paper to enhance the flow and improve the language.

Response:

Thank you for your valuable feedback. We will carefully revise the manuscript to improve its flow and enhance the language. We understand the importance of clear and concise communication in academic writing, and we will make the necessary adjustments to ensure that the paper meets the highest standards of readability and coherence.

Reviewer#5 Comment 4: Several key works are not mentioned, even they are highly related to this work. For example, “Hypernetwork-based physics-driven personalized federated learning for CT imaging”, “Physics-Driven Spectrum-Consistent Federated Learning for Palmprint Verification” and “FCSCNN: Feature centralized Siamese CNN-based android malware identification”.

Response:

Thank you for pointing this out. We have added citations for these articles in the Introduction section of the paper, as referenced in [12-14].

Reviewer#5 Comment 5: Please enhance the image quality.

Response:

Thank you for your suggestion. We will improve the image quality in the revised version of the paper to ensure that all visual elements are clear and of high resolution.

Reviewer#5 Comment 6: The equation is not well described. For example, in Eq. 1, i is the webpage, this variable should be the subscript. I do believe the input of PR is not only the index, right?

Response:

Thank you for pointing this out. In response to your concerns, we have made revisions to the manuscript, such as in Equation 11. To better align with the content of our research, the original variable has been changed to , and the meanings of the other parameters have been adjusted accordingly.

Reviewer#5 Comment 7: The formula format is not standardized. When the formula variables are not clearly explained, a comma should follow the formula, and a "where" should be placed below it. The variables should then be explained within a fixed frame.

Response:

Thank you for your feedback. We have reviewed the formula formatting throughout the manuscript and made revisions according to your suggestions. We hope that the updated format meets the requirements.

Reviewer#5 Comment 8: Why do you compare methods that are different in different datasets? Can you reproduce these methods following their papers?

Response:

Thank you for pointing this out.In Table 2, we compare our method with MLP, N-gram, LSTM, and SVM on two datasets. These four methods were implemented in our code, and applying them to both datasets was done to enhance the generalization ability of our algorithm. Table 3 presents a comparison with algorithms from other papers. The methods SDGNet, Mal-ASSF, and LGMal all used Dataset 1, so we directly used the data from those papers for comparison. Catak et al. [50] used Dataset 2, so we also used the data from their paper for comparison.

Reviewer#5 Comment 9: As you said, SMOTE is designed to alleviate the unbalanced problem. The key ablation study about this is missing, and please show the results w/ and w/o SMOTE following Fig. 11.

Response:

Thank you for your feedback. In Table 6 of the Ablation Studies subsection within the Results and Discussion section, we conducted an analysis w/ and w/o SMOTE

Reviewer#5 Comment 10: Besides, I think SMOTE must lead to an overfitting problem, please discuss it. I think this problem is very important, unless you can discuss it sufficiently, or this method is not convincing.

Response:

Thank you for pointing out the issue. Based on your observation, we found that there was indeed overfitting in the comparison of the loss functions between w/o SMOTE and w/ SMOTE (originally Fig 13). To address this, we increased the regularization strength in the CNN model, and the overfitting issue has now been significantly improved, as shown in Fig 10.

We trust that these responses and the associated revisions thoroughly address your concerns and enhance the manuscript's quality. We are grateful for the opportunity to refine our work and eagerly await any further feedback.

Best regards,

[Cuihua Ma,Zhenwan Li,Haixia Long,Anas Bilal,Xiaowen Liu]

---

## [Decision Letter · Decision Letter 3]

6 Dec 2024

PONE-D-23-26475R3A Malware Classification Method Based on Directed API Call RelationshipsPLOS ONE

Dear Dr. Bilal,

Thank you for submitting your manuscript to PLOS ONE. After careful consideration, we feel that it has merit but does not fully meet PLOS ONE’s publication criteria as it currently stands. Therefore, we invite you to submit a revised version of the manuscript that addresses the points raised during the review process.

We look forward to receiving your revised manuscript.

Kind regards,

Hikmat Ullah Khan, PhD (Computer Science)

Academic Editor

PLOS ONE

**Journal Requirements:**

Reviewers' comments:

Reviewer's Responses to Questions

**Comments to the Author**

1. If the authors have adequately addressed your comments raised in a previous round of review and you feel that this manuscript is now acceptable for publication, you may indicate that here to bypass the “Comments to the Author” section, enter your conflict of interest statement in the “Confidential to Editor” section, and submit your "Accept" recommendation.

Reviewer #4: All comments have been addressed

Reviewer #5: All comments have been addressed

Reviewer #6: (No Response)

Reviewer #7: All comments have been addressed

2. Is the manuscript technically sound, and do the data support the conclusions?

Reviewer #4: Yes

Reviewer #5: Yes

Reviewer #6: Yes

Reviewer #7: Yes

3. Has the statistical analysis been performed appropriately and rigorously? 

Reviewer #4: Yes

Reviewer #5: Yes

Reviewer #6: Yes

Reviewer #7: Yes

4. Have the authors made all data underlying the findings in their manuscript fully available?

Reviewer #4: Yes

Reviewer #5: Yes

Reviewer #6: No

Reviewer #7: Yes

5. Is the manuscript presented in an intelligible fashion and written in standard English?

Reviewer #4: Yes

Reviewer #5: Yes

Reviewer #6: Yes

Reviewer #7: Yes

6. Review Comments to the Author

**Reviewer #4: **All problems have been addressed. I have not extra problems. All problems have been addressed. I have not extra problems.

**Reviewer #5: **The authors have well addressed my concerns, I think this version can be accepted.

**Reviewer #6:** Overall paper is well structured. However, address following points for further improvements.

• The abstract doesn’t address how the method handles real-world scenarios like noisy or imbalanced datasets.

• The introduction briefly mentions real-world scenarios but could benefit from elaborating on how the method handles challenges like imbalanced datasets or novel malware does not present in training data.

• Ensure consistent use of terms like "graph convolutional networks (GCN)" and "directed graph convolutional networks (DGCN)" across sections to avoid confusion.

• Some references, such as those for malware statistics, could be better integrated into the discussion to highlight their relevance to the problem statement.

• Ensure all abbreviations like FSADGCN, CNN, and API are defined explicitly when first introduced. For example, "API sequence instructions" can be expanded to "Application Programming Interface (API) sequence instructions."

• Replace the placeholder text "Fig 1 venenatis sed ipsum varius..." with an appropriate description or remove unrelated filler text. Ensure figure references are meaningful and properly aligned with the content.

• Rephrase sentences to improve readability, such as changing "The extracted node feature attributes are then subjected to..." to "Next, the extracted node feature attributes undergo..."

• Review and adjust the formatting of equations to ensure clarity and consistency, such as spacing and alignment in equations (e.g., Equations 1-11). Use inline or displayed math consistently to improve readability.

**Reviewer #7:** In this study, the authors have proposed a malware classification method using directed graphs of API call sequences, processed with first- and second-order graph convolutional networks (FSGCN). The embeddings are converted to grayscale images for CNN classification, with SMOTE addressing dataset imbalance. The authors have conducted an ablation study and provided a satisfactory analysis of the proposed approach. However, following suggestions can further enhance the study:

i. Clearly mention the improvements in the abstract section.

ii. The image quality of Figure 4 needs improvement as the text is unclear, and the color scheme for the bars could be enhanced for better visual distinction.

iii. Presenting the dataset description in a tabular format in section could more effectively showcase the dataset statistics.

7. PLOS authors have the option to publish the peer review history of their article (what does this mean?). If published, this will include your full peer review and any attached files.

Reviewer #4: No

Reviewer #5: No

Reviewer #6: No

Reviewer #7: No

---

## [Author Response · Author response to Decision Letter 4]

11 Dec 2024

Dear Reviewers,

Thank you for your thorough review and valuable feedback on our manuscript "Malware classification through API Calls multi-dimensional feature extrate based on SMOTE and Directed Graph convolution network and CNN". We have carefully considered your comments and made the following revisions to our manuscript:

Reviewer#6 Comment 1:The abstract doesn’t address how the method handles real-world scenarios like noisy or imbalanced datasets.

Response:

We appreciate your comment about real-world challenges such as noisy or imbalanced datasets. In the revised abstract, we have addressed the handling of imbalanced datasets by incorporating the Synthetic Minority Over-sampling Technique (SMOTE), which ensures that underrepresented classes are sufficiently represented during training. a

Reviewer#6 Comment 2: The introduction briefly mentions real-world scenarios but could benefit from elaborating on how the method handles challenges like imbalanced datasets or novel malware does not present in training data.

Response:

Thank you for your insightful feedback. We appreciate the suggestion to elaborate on how our proposed method handles real-world challenges such as imbalanced datasets and novel malware variants that are not present in the training data. In response to your comment, we have revised the relevant sections of the introduction to provide a clearer explanation of how our method addresses these challenges.

Modifications in the Introduction:

Class Imbalance: We have explicitly addressed the issue of class imbalance, a common challenge in real-world malware classification. In the revised text, we highlight how our approach leverages the Synthetic Minority Over-sampling Technique (SMOTE), a widely used technique for balancing datasets by generating synthetic samples for underrepresented classes. By ensuring a more balanced dataset, SMOTE helps prevent the model from being biased towards the majority class, which in turn improves the model's ability to detect less frequent malware types, including rare or emerging ones.

Novel Malware Detection: We also discuss the challenge of detecting novel malware variants—those that were not observed during training. Our approach's use of a directed graph representation of API call sequences provides a more robust and flexible method compared to traditional static feature-based approaches. This graph-based method better captures the structural and sequential relationships in malware behavior, allowing the model to generalize more effectively to unseen malware variants. The flexibility of our approach enhances its ability to detect novel malware, even those that exhibit behavior outside of the training data distribution.

Reviewer#6 Comment 3: Ensure consistent use of terms like "graph convolutional networks (GCN)" and "directed graph convolutional networks (DGCN)" across sections to avoid confusion.

Response:

Thank you very much for your valuable feedback and for pointing out the importance of consistent terminology usage within our manuscript. We acknowledge that maintaining consistency in the terms used is crucial for clarity and to avoid any potential confusion for the readers.

In response to your comment, we have carefully reviewed the entire manuscript and ensured that the terms "graph convolutional networks (GCN)" and "directed graph convolutional networks (DGCN)" are used consistently across all sections. Specifically, we have:

(1) Conducted a thorough search for all instances where these terms appear.

Verified that the full form is introduced before or at the first mention of the acronym in the text.

(2) Corrected any variations in the usage of these terms to ensure uniformity.

(3) Checked for any potential ambiguity and clarified the context where necessary.

We believe these changes will enhance the readability and precision of our paper. We appreciate your attention to this detail and hope that our revisions meet with your approval.

Reviewer#6 Comment 4: Some references, such as those for malware statistics, could be better integrated into the discussion to highlight their relevance to the problem statement.

Response:

We greatly appreciate your thoughtful feedback, especially regarding the integration of malware statistics and the importance of clearly articulating the relevance of these figures to the research problem. In response, we have made several changes to the introduction to better align these statistics with the challenges in malware detection.

(1) Integration of Malware Statistics: As per your suggestion, we have enhanced the connection between the provided malware statistics (such as the increase in ransomware incidents and overall malware activity) and the growing complexity of malware threats. We now explicitly highlight how these statistics underscore the escalating volume and sophistication of malware, directly motivating the need for more advanced malware detection methods. For example, we now emphasize that the 13% year-on-year increase in ransomware incidents (according to the Verizon Business 2022 Data Breach Investigations Report) highlights an urgent need for effective and adaptive malware detection techniques.

(2) Clearer Articulation of the Research Problem: The discussion now more clearly emphasizes how the growing diversity and sophistication of malware types make traditional detection methods insufficient. We connect these challenges to the limitations of static and traditional machine learning methods, making a stronger case for the need for more sophisticated, dynamic techniques—such as our novel approach using directed graph representations of API call sequences.

We believe these revisions will provide a more coherent narrative that effectively ties together the problem context, the limitations of existing approaches, and the motivation for our proposed solution. Thank you again for your valuable input, and we look forward to your further feedback.

Reviewer#6 Comment 5: Ensure all abbreviations like FSADGCN, CNN, and API are defined explicitly when first introduced. For example, "API sequence instructions" can be expanded to "Application Programming Interface (API) sequence instructions."

Response:

Thank you for your meticulous review and the valuable suggestion to ensure that all abbreviations are explicitly defined upon their first introduction in the manuscript. We agree that this practice enhances the clarity and accessibility of our paper, especially for readers who may not be familiar with the specific terminologies used.

In response to your comment, we have made the following adjustments:

(1) Explicit Definition of Abbreviations: We have reviewed the entire document and identified all instances where abbreviations such as FSADGCN, CNN, and API are used. For each abbreviation, we have provided a full expansion at its first appearance in the text. For example, "API sequence instructions" has been revised to "Application Programming Interface (API) sequence instructions."

(2) Consistency Check: In addition to defining abbreviations, we have ensured that they are used consistently throughout the manuscript. This means that once an abbreviation is introduced, it is used in subsequent mentions, unless the context requires the full term for clarity.

(3) Verification: To prevent any oversight, we have conducted a final verification to confirm that no abbreviations are left undefined and that all expansions are accurate and appropriate.

We believe these changes will significantly improve the clarity and comprehensibility of our work. We appreciate your attention to detail and thank you for your constructive feedback.

Reviewer#6 Comment 6: Replace the placeholder text "Fig 1 venenatis sed ipsum varius..." with an appropriate description or remove unrelated filler text. Ensure figure references are meaningful and properly aligned with the content.

Response:

Thank you for your valuable feedback regarding the placeholder text in Figure 1 and the importance of meaningful figure references. We appreciate your attention to detail, which has helped us improve the clarity and coherence of our manuscript. In response to your comment, we have made the following revisions:

(1) Replacement of Placeholder Text:

We have replaced the placeholder text "Fig 1 venenatis sed ipsum varius..." with a clear and detailed description of Figure 1. The new caption now accurately reflects the content and purpose of the figure, providing readers with a comprehensive understanding of the illustrated process.

(2) Enhanced Figure Description:

For Figure 1, the updated caption reads: "Workflow of the proposed malware classification method based on FSGCN. This figure illustrates the process from handling imbalanced datasets using the SMOTE algorithm, constructing directed graphs for API sequences, extracting multi-dimensional attribute features, computing first-order and second-order adjacency matrices, generating feature embeddings with GCN, converting these embeddings into grayscale images, and finally classifying the images using CNN."

We believe these changes will significantly enhance the clarity and impact of our paper. We appreciate your constructive comments and hope that the revised manuscript meets with your approval.

Reviewer#6 Comment 7: Rephrase sentences to improve readability, such as changing "The extracted node feature attributes are then subjected to..." to "Next, the extracted node feature attributes undergo..."

Response:

Thank you for your valuable feedback. We have carefully considered your suggestion to improve sentence readability, particularly the one regarding "The extracted node feature attributes are then subjected to...". We agree that using more direct and concise language improves clarity and flow. As recommended, we have rephrased this sentence to "Next, the extracted node feature attributes undergo...," which simplifies the construction and enhances readability without altering the meaning.

Additionally, we have applied similar revisions throughout the paper to ensure a consistent and formal tone, enhancing overall readability and making the text more accessible to readers. We appreciate your insights and believe these adjustments improve the quality of the manuscript.

Reviewer#6 Comment 8: Review and adjust the formatting of equations to ensure clarity and consistency, such as spacing and alignment in equations (e.g., Equations 1-11). Use inline or displayed math consistently to improve readability.

Response:

Thank you for your thoughtful review. We appreciate your recommendation to review and adjust the formatting of the equations to improve clarity and consistency. We understand the importance of clear and well-aligned mathematical expressions, and we have made the necessary revisions to ensure that all equations (e.g., Equations 1-11) are properly spaced and aligned.

As per your suggestion, we have ensured that:

(1) Spacing: We have adjusted the spacing within and around the equations to ensure readability and consistency. For instance, we have ensured consistent use of spacing between variables, operators, and functions.

(2) Alignment: Equations that span multiple lines have been aligned correctly to enhance visual clarity, making them easier to follow.

(3) Math Formatting: We have reviewed our use of inline and displayed math expressions and ensured they are used consistently throughout the paper. Displayed equations are now reserved for more complex expressions, while inline math is used for simpler notations to improve readability and flow.

Since we used the LaTeX template provided by the journal, we have made sure that the formatting adheres to the journal's guidelines while implementing the adjustments you suggested.

Thank you again for your helpful feedback, which we believe enhances the overall presentation and readability of our manuscript.

Reviewer#7 Comment 1: Clearly mention the improvements in the abstract section.

Response:

Thank you for your valuable feedback. We appreciate your suggestion to clearly highlight the improvements in the abstract section. In response to your comment, we have revised the abstract to explicitly emphasize the key advancements and contributions of our work. The updated abstract now clearly outlines the innovations in our approach, particularly focusing on how our method addresses the limitations of existing malware classification techniques by incorporating directed API call relationships, utilizing GCN for feature extraction, and applying the SMOTE algorithm for handling imbalanced datasets.

By highlighting these improvements, we aim to provide readers with a concise and clear understanding of the novel contributions of our research right from the abstract.

Thank you again for your insightful recommendation, which has helped us refine the presentation of our work.

Reviewer#7 Comment 2: The image quality of Figure 4 needs improvement as the text is unclear, and the color scheme for the bars could be enhanced for better visual distinction.

Response:

Thank you for your detailed feedback on the image quality of Figure 4 and the suggestions for improving its visual presentation. We fully agree that the clarity of the text and the color scheme are critical for effective communication of our findings.

In response to your comments, we have taken the following actions:

(1) Image Quality Enhancement: We have reprocessed Figure 4 using higher resolution graphics to ensure that all textual elements are clear and legible. This should improve the overall readability of the figure.

(2) Color Scheme Optimization: We have revised the color scheme for the bars in Figure 4 to provide better visual distinction between different categories. We have chosen a set of colors that not only enhance contrast but also adhere to color accessibility guidelines, ensuring that the figure is accessible to all readers。

We believe these changes will significantly improve the interpretability and visual appeal of Figure 4, thereby enhancing the overall quality of the manuscript. Your constructive criticism has been invaluable in helping us refine our work.

Reviewer#7 Comment 3: Presenting the dataset description in a tabular format in section could more effectively showcase the dataset statistics.

Response:

Thank you for your valuable feedback regarding the presentation of the dataset description. We agree that showcasing the dataset statistics in a tabular format within the relevant section can enhance the clarity and accessibility of the information for readers.

In response to your suggestion, we have revised the manuscript to include a detailed table (Table 2) that encapsulates the overall situation and key characteristics of the datasets. This adjustment aims to provide a more structured overview, allowing for a more effective comparison and understanding of the data used in our study.

We believe this change significantly improves the presentation of our dataset and aids in better communicating our research to the academic community. Your insight has been instrumental in refining our work.

We trust that these responses and the associated revisions thoroughly address your concerns and enhance the manuscript's quality. We are grateful for the opportunity to refine our work and eagerly await any further feedback.

Best regards,

[Cuihua Ma,Zhenwan Li,Haixia Long,Anas Bilal,Xiaowen Liu]

---

## [Decision Letter · Decision Letter 4]

20 Dec 2024

A Malware Classification Method Based on Directed API Call Relationships

PONE-D-23-26475R4

Dear Dr. Bilal,

We’re pleased to inform you that your manuscript has been judged scientifically suitable for publication and will be formally accepted for publication once it meets all outstanding technical requirements.

Kind regards,

Hikmat Ullah Khan, PhD (Computer Science)

Academic Editor

PLOS ONE

Additional Editor Comments (optional):

Reviewers' comments:

Reviewer's Responses to Questions

**Comments to the Author**

1. If the authors have adequately addressed your comments raised in a previous round of review and you feel that this manuscript is now acceptable for publication, you may indicate that here to bypass the “Comments to the Author” section, enter your conflict of interest statement in the “Confidential to Editor” section, and submit your "Accept" recommendation.

Reviewer #7: All comments have been addressed

2. Is the manuscript technically sound, and do the data support the conclusions?

Reviewer #7: Yes

3. Has the statistical analysis been performed appropriately and rigorously? 

Reviewer #7: Yes

4. Have the authors made all data underlying the findings in their manuscript fully available?

Reviewer #7: Yes

5. Is the manuscript presented in an intelligible fashion and written in standard English?

Reviewer #7: Yes

6. Review Comments to the Author

Reviewer #7: The author has addressed all my comments based on revision 3. Revision 4 is updated and well-suited for publishing.

7. PLOS authors have the option to publish the peer review history of their article (what does this mean?). If published, this will include your full peer review and any attached files.

Reviewer #7: No

---

## [Editor Report · Acceptance letter]

PONE-D-23-26475R4

PLOS ONE

Dear Dr. Bilal,

I'm pleased to inform you that your manuscript has been deemed suitable for publication in PLOS ONE. Congratulations! Your manuscript is now being handed over to our production team.

Kind regards,

on behalf of

Dr. Hikmat Ullah Khan

Academic Editor

PLOS ONE